# ImageBART: Bidirectional Context with Multinomial Diffusion for Autoregressive Image Synthesis

**Patrick Esser**[*]     **Robin Rombach**[*]     **Andreas Blattmann**[*]     **Björn Ommer**
Ludwig Maximilian University of Munich & IWR, Heidelberg University, Germany
`https://compvis.github.io/imagebart/`

## Abstract

Autoregressive models and their sequential factorization of the data likelihood have recently demonstrated great potential for image representation and synthesis. Nevertheless, they incorporate image context in a linear 1D order by attending only to previously synthesized image patches above or to the left. Not only is this unidirectional, sequential bias of attention unnatural for images as it disregards large parts of a scene until synthesis is almost complete. It also processes the entire image on a single scale, thus ignoring more global contextual information up to the gist of the entire scene. As a remedy we incorporate a coarse-to-fine hierarchy of context by combining the autoregressive formulation with a multinomial diffusion process: Whereas a multistage diffusion process successively removes information to coarsen an image, we train a (short) Markov chain to invert this process. In each stage, the resulting autoregressive ImageBART model progressively incorporates context from previous stages in a coarse-to-fine manner. Experiments show greatly improved image modification capabilities over autoregressive models while also providing high-fidelity image generation, both of which are enabled through efficient training in a compressed latent space. Specifically, our approach can take unrestricted, user-provided masks into account to perform local image editing. Thus, in contrast to pure autoregressive models, it can solve free-form image inpainting and, in the case of conditional models, local, text-guided image modification without requiring mask-specific training.

## 1  Introduction

Spurred by the increasingly popular attention mechanism, a remarkably simple principle has driven progress in deep generative modeling over the past few years: Factorizing the likelihood of the data in an autoregressive (AR) fashion

$$p(x) = \prod_i p_\theta(x_i|x_{<i}) \tag{1}$$

and subsequently learning the conditional transition probabilities with an expressive neural network such as a transformer [75]. The success of this approach is evident in domains as diverse as language modeling [7], music generation [16], neural machine translation [46, 76], and (conditional) image synthesis [54, 8]. However, especially for the latter task of image synthesis, which is also the focus of this work, the high dimensionality and redundancy present in the data challenges the direct applicability of this approach.

**Missing Bidirectional Context** Autoregressive models which represent images as a sequence from the top-left to the bottom-right have demonstrated impressive performance in sampling novel images and completing the lower half of a given image [8, 21]. However, the unidirectional, fixed ordering

---

[*]The first three authors contributed equally to this work.

35th Conference on Neural Information Processing Systems (NeurIPS 2021).

of sequence elements not only imposes a perceptually unnatural bias to attention in images by only considering context information from left or above. It also limits practical applicability to image modification: Imagine that you only have the lower half of an image and are looking for a completion of the upper half then these models fail at this minor variation of the completion task. The importance of contextual information from both directions [36] has also been recognized in the context of language modeling [14, 45]. However, simply allowing bidirectional context as in [14] does not provide a valid factorization of the density function for a generative model. Furthermore, the sequential sampling strategy introduces a gap between training and inference, as training relies on so-called teacher-forcing [3] (where ground truth is provided for each step) and inference is performed on previously sampled tokens. This *exposure bias* can introduce significant accumulations of errors during the generation process, affecting sample quality and coherence [57].

**Global Context & Control via Multinomial Diffusion** We propose a coarse-to-fine approach that addresses the unidirectional bias of generative autoregressive models and their exposure bias as well as the lacking global context. We formulate learning the data density as a hierarchical problem. A coarser stage provides compressed contextual side information about the *entire* image for the autoregressive process on the next finer stage. We utilize a diffusion process to gradually eliminate information and compress the data, yielding a hierarchy of increasingly abstract and compact representations. The first scale of this approach is a discrete representation learning task (cf. [74, 58, 16, 21, 78, 56]). Subsequently, we further compress this learned representation via a fixed, multinomial diffusion process [65, 30]. We then invert this process by training a Markov chain to recover the data from this hierarchy. Each Markovian transition is modeled autoregressively but it simultaneously attends to the preceding state in the hierarchy, which provides crucial global context to each individual autoregressive step. As each of this steps can also be interpreted as learning a denoising cloze task [45], where missing tokens at the next finer stage are "refilled" with a bidirectional encoder and an autoregressive decoder, we dub our approach *ImageBART*.

**Contributions of our work** Our approach tackles high-fidelity image synthesis with autoregressive models by learning to invert a fixed multinomial diffusion process in a discrete space of compact image representations to successively introduce context. This reduces both the often encountered exposure bias of AR models and also enables locally controlled, user-interactive image editing. Additionally, our model effectively handles a variety of conditional synthesis tasks and our introduced hierarchy corresponds to a successively compressed image representation. We observe that our model sample visually plausible images while still enabling a trade-off between reconstruction capability and compression rate.

## 2   Related Work

**Latent Variable Models** Among likelihood-based approaches, latent variable models represent a data distribution with the help of unobserved latent variables. For example, Variational Autoencoders (VAEs) [38, 59] encode data points into a lower dimensional latent variable with a factorized distribution. This makes them easy to sample, interpolate [44, 37] and modify [77]. In a conditional setting [39], latent variables which are independent from the conditioning lead to disentangled representations [31, 69, 48, 60, 5]. A hierarchy of latent variables [66] gives mutli-scale representations of the data. Unfortunately, even the deepest instantiations of these models [47, 71, 10] lack in sample quality compared to other generative models and are oftentimes restricted to highly regular datasets.

**Autoregressive Models** AR models represent a distribution as a product of conditional, learnable factors via the chain rule of probability densities. While this makes them powerful models for density estimation [70, 24], their samples often lack global consistency. Especially on image data modeled with convolutional architectures [73, 62], this has been attributed to a locality bias of convolutional neural networks (CNNs) which biases the model towards strong local correlations between neighboring pixels at the expense of a proper modeling of coherence [40, 22]. This leads to samples resembling texture patterns without discernible global structure. Attempts to fix this properties by including explicit latent variables [27, 9, 22] have not been overly successful, mainly due the expressiveness of AR models, providing little incentive for learning additional latent variables.

**Generative Models on Improved Representations** Another successful line of work first learn an improved image representation and subsequently learn a generative model for this representation [74, 12]. Most works [58, 21, 56] learn a discrete representation which is subsequently modeled autoregressively but approaches using continuous representations in combination with VAEs [12], or

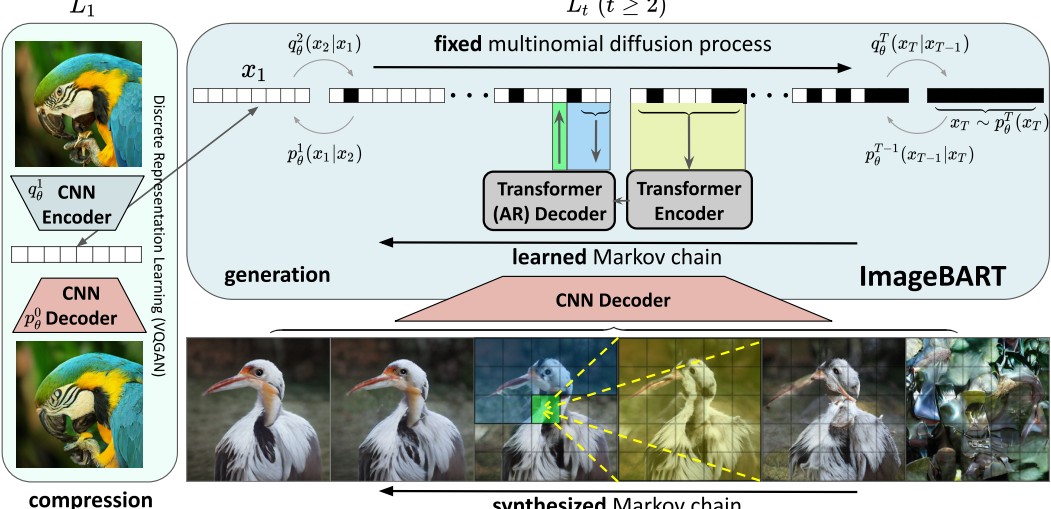

Figure 1: Overview over our approach: We first learn a compressed, discrete image representation $x_1$ and subsequently our generative ImageBART model reverts a fixed multinomial diffusion process via a Markov Chain, where the individual transition probabilities are modeled as independent autoregressive encoder-decoder models. This introduces a coarse-to-fine hierarchy such that each individual AR model can attend to global context from its preceding scale in the hierarchy.

normalizing flows [1, 60, 20, 4, 18], exist too. Learning a compact representation enables the use of transformers for autoregressive modeling [8], which avoids the locality bias of CNNs, can be used for the synthesis of complex scenes conditioned on text as in DALL-E [56], and, when combined with adversarial learning [25], enables sampling of coherent high-resolution images [21]. However, AR modeling of a learned representation still limits applications compared to latent variable models. Their samples can still exert artifacts resulting from a sequential modeling of components, and, since these models are always trained by "teacher-forcing", they are susceptible to an exposure bias [3, 57, 26, 63, 43].

**Diffusion Probabilistic Models** Diffusion probabilistic models revert a fixed, diffusion process with a learned Markov Chain [65]. Being directly applied in pixel space, however, downstream analysis reveals that these models tend to optimize subtle details of the modeled data, which have little contribution to the sample quality [29, 15], particularly hindering applications on high-resolution and -complexity datasets. By using a multinomial diffusion process [30] (recently generalized by [2]) on a compressed, discrete representation of images, we circumvent these issues. Diffusion probabilistic models require a very large number of diffusion steps in order to model the reverse process with a model distribution that factorizes over components. Because our approach uses autoregressively factorized models for the reverse process, we can reduce the required number of steps and obtain significant improvements in sampling speed and the ability to model complex datasets.

## 3 Method

### 3.1 Hierarchical Generative Models

To tackle the difficult problem of modeling a highly complex distribution $p(x)$ of high-dimensional images $x$, we (i) introduce bidirectional context into an otherwise unidirectional autoregressive factorization of $p(x)$ as in Eq. (1) and (ii) reduce the difficulty of the learning problem with a hierarchical approach. To do so, we learn a sequence of distributions $(p_\theta^t)_{t=0}^T$, such that each distribution $p_\theta^{t-1}$ models a slightly more complex distribution with the help of a slightly simpler distribution $p_\theta^t$ one level above. This introduces a coarse-to-fine hierarchy of image representations $x_{0:T} := (x_t)_{t=0}^T$, such that an $x_{t-1}$ is modeled conditioned on $x_t$, i.e. $x_{t-1} \sim p_\theta^{t-1}(x_{t-1}|x_t)$ and defines a reverse Markov Chain for $x =: x_0$ as $p_\theta(x_0) = p_\theta^T(x_T) \prod_{t=1}^T p_\theta^{t-1}(x_{t-1}|x_t)$. Since our goal is to approximate the original distribution $p(x)$ with $p_\theta(x_0)$, we introduce a forward Markov Chain, $q_\theta(x_{1:T}|x_0) = \prod_{t=1}^T q_\theta^t(x_t|x_{t-1})$, to obtain a tractable upper bound on the Kullback-Leibler (KL) divergence between $p$ and $p_\theta$, $\mathbb{KL}(p(x_0)\|p_\theta(x_0)) =: \mathcal{KL}$, using the evidence lower bound

(ELBO). With $q_\theta^T(x_T|x_{T-1}) := p_\theta^T(x_T)$, we obtain

$$\mathcal{KL} \leq \underbrace{\mathbb{E}_{x_0,x_1} \log \frac{p(x_0)}{p_\theta^0(x_0|x_1)}}_{=:L_1 \to \text{ discrete repr. learning}} + \sum_{t=2}^{T} \underbrace{\mathbb{E}_{x_0,x_t} \mathbb{KL}(q_\theta^{t-1}(x_{t-1}|x_t,x_0)\|p_\theta^{t-1}(x_{t-1}|x_t))}_{=:L_t \to \text{ decoupled with diffusion process}} \quad (2)$$

We use $L_1$ to learn a compressed and discrete representation of images, such that subsequent stages of the hierarchy do not need to model redundant information (Sec. 3.2). With $L_t, t > 1$ we learn a model that can rely on global context from a coarser representation $x_t$ to model the representation $x_{t-1}$ (Sec. 3.3). See Fig. 1 for an overview of the proposed model.

## 3.2 Learning a compact, discrete representation for images

Since the first stage of the hierarchical process is the one that operates directly on the data, we assign it a separate role. To avoid that the optimization of $L_t$ $(t = 1, \ldots, T)$ in Eq. (2) unnecessarily wastes capacity on redundant details in the input images—which is an often encountered property of pixel-based likelihood models [74, 21, 50]—we take $L_1 = \mathbb{E}_{p(x_0)q_\theta^1(x_1|x_0)} \log \frac{p(x_0)}{p_\theta^0(x_0|x_1)}$ to be the reconstruction term for a discrete autoencoder model. This has the advantage that we can directly build on work in neural discrete representation learning, which has impressively demonstrated that discrete representations can be used for high-quality synthesis of diverse images while achieving strong compression. In particular, [49] and [21] have shown that adding an adversarial realism prior to the usual autoencoder objective helps to produce more realistic images at higher compression rates by locally trading reconstruction fidelity for realism.

More specifically, we follow [21] to encode images into a low-dimensional representation which is then vector-quantized with a learned codebook of size $K$ to obtain $\{0, \ldots, K-1\}^{h \times w} \ni x_1 \sim q_\theta^1(x_1|x_0)$ deterministically as the index of the closest codebook entry. The encoder is a convolutional neural network (CNN) with four downsampling steps, such that $h = H/16$ and $w = W/16$ for any input image $x_0 \in \mathbb{R}^{H \times W \times 3}$. For downstream autoregressive learning, this representation is then unrolled into a discrete sequence of length $N = h \cdot w$. To recover an image from $x_1$, we utilize a CNN decoder $G$, such that the reverse model is specified as

$$-\log p_\theta^0(x_0|x_1) \propto f_{rec}(x_0, G_\theta(x_1)) + \log D_\phi(G_\theta(x_1)) =: L_{rec}(x_0, x_1; \theta) + L_{adv}(x_1; \theta, \phi) \quad (3)$$

Here, $f_{rec}$ denotes the perceptual similarity metric [23, 32, 19, 80] (known as LPIPS) and $D_\phi$ denotes a patch-based adversarial discriminator [25]. Note that, due to the deterministic training, the likelihood in Eq. (3) is likely to be degenerate. $D_\phi$ is optimized to differentiate original images $x_0$ from their reconstruction $G_\theta(x_1)$ using simultaneous gradient ascent, such that the objective for learning the optimal parameters $\{\theta^*, \phi^*\}$ reads:

$$\{\theta^*, \phi^*\} = \arg\min_\theta \max_\phi \left( L_{rec}(x_0, x_1; \theta) - L_{adv}(x_1; \theta, \phi) + \log D_\phi(x_0) + L_{cb}(\theta) \right) \quad (4)$$

The optimization of $\theta$ via this objective includes the parameters of the encoder and decoder in addition to the parameters of the learned codebook, trained via the codebook loss $L_{cb}$ as in [74, 21].

## 3.3 Parallel learning of hierarchies

Under suitable choices for $p_\theta, q_\theta$, one can directly optimize these chains over $\sum_t L_t$. However, the objectives $L_t$ of the hierarchy levels are coupled through the forward chain $q_\theta$, which makes this optimization problem difficult. With expressive reverse models $p_\theta^{t-1}$, the latent variables $x_t$ are often ignored by the model [22] and the scale of the different level-objectives can be vastly different, resulting in a lot of gradient noise that hinders the optimization [52]. In the continuous case, reweighting schemes for the objective can be derived [29] based on a connection to score matching models [67]. However, since we are working with a discrete $x_1$, there is no analogue available.

While we could follow the approach taken for the first level and sequentially optimize over the objectives $L_t$, this is a rather slow process since each level $t-1$ needs to be converged before we can start solving level $t$. However, this sequential dependence is only introduced through the forward models $q_\theta^t$ and since $q_\theta^1$ already learns a strong representation, we can choose simpler and fixed, predefined forward processes for $q_\theta^t, t > 1$. The goal of these processes, i.e., generating a hierarchy of distributions by reducing information in each transition, can be readily achieved by, e.g., randomly masking [14], removing [45] or replacing [30] a fraction of the components of $x_{t-1}$.

**Multinomial diffusion**  This process of randomly replacing a fraction $\beta_t$ of the components with random entries can be described as a multinomial diffusion process [30], a natural generalization of binomial diffusion [65]. The only parameter $\theta$ of $q_\theta^t$ is therefore $\beta_t$, which we consider to be fixed. Using the standard basis $e(k) = (\delta_{jk})_{j=1}^K$, the forward process can be written as a product of categorical distributions $\mathcal{C}$ specified in terms of the probabilities over the codebook indices:

$$q_\theta^t(x_t|x_{t-1}) = \prod_{i=1}^N \mathcal{C}(x_t^i|(1-\beta_t)e(x_{t-1}^i) + \beta_t \mathbb{1}/K), \quad t > 1 \tag{5}$$

where $\mathbb{1} = (1)_{j=1}^K$ is the all one vector. It then follows that after $t-1$ steps, on average, a fraction of $\bar{\alpha}_t := \prod_{l=2}^t (1-\beta_t)$ entries from $x_1$ remain unchanged in $x_t$, i.e.

$$q_\theta^t(x_t|x_1) = \prod_{i=1}^N \mathcal{C}(x_t^i|\bar{\alpha}_t e(x_1^i) + (1-\bar{\alpha}_t)\mathbb{1}/K), \quad t > 1. \tag{6}$$

This enables computation of the posterior $q_\theta(x_{t-1}|x_t, x_1) = \frac{q_\theta^t(x_t|x_{t-1})q_\theta(x_{t-1}|x_1)}{q_\theta(x_t|x_1)}$ for $t > 2$, and, using the fact that $q_\theta^1$ is deterministic, we can rewrite $L_t$ as

$$\mathbb{E}_{p(x_0)}\mathbb{E}_{q_\theta(x_t|x_1)}\mathbb{KL}(q_\theta^{t-1}(x_{t-1}|x_t, x_1)\|p_\theta^{t-1}(x_{t-1}|x_t)), \quad t > 2 \tag{7}$$

such that the KL term can now be computed analytically for $t > 2$. For $t = 2$, we use a single sample Monte-Carlo estimate for the maximum likelihood reformulation, i.e.

$$\arg\min L_2 = \arg\max \mathbb{E}_{p(x_0)}\mathbb{E}_{q_\theta^2(x_2|x_1)} \log p_\theta^1(x_1|x_2). \tag{8}$$

Finally, we set $p_\theta^T$ to be a uniform distribution. This completes the definition of the reverse chain $p_\theta$, which can now be started from a random sample for $x_T \sim p_\theta^T(x_T)$, denoised sequentially through $x_{t-1} \sim p_\theta^{t-1}(x_{t-1}|x_t)$ for $t = T, \ldots, 2$, and finally be decoded to a data sample $x_0 = G(x_1)$.

**Reverse diffusion models**  Under what conditions can we recover the true data distribution? By rewriting $\sum_t L_t$, we can see from

$$\mathbb{KL}(p(x_0)\|p_\theta(x_0)) \le \sum_{t=1}^T \mathbb{KL}(q_\theta(x_{t-1}|x_t)\|p_\theta^{t-1}(x_{t-1}|x_t)) \tag{9}$$

that this is possible as long as all reverse models are expressive enough to represent the true reverse processes defined by $q_\theta$. For the first level, we can ensure this by making $x_1$ large enough such that the reconstruction error becomes negligible. For the diffusion process, previous image models [65, 29, 68, 30] relied on the fact that, in the limit $\beta_t \to 0$, the form of the true reverse process has the same functional form as the forward diffusion process [65, 41]. In particular, this allows modeling of the reverse process with a distribution factorized over the components. However, to make $q_\theta^{T-1}$ close to a uniform distribution requires a very large $T$ (in the order of 1000 steps) with small $\beta_t$. Training such a large number of reverse models is only feasible with shared weights for the models, but this requires a delicate reweighting [29] of the objective and currently no suitable reweighting is known for the discrete case considered here.

Thus, to be able to recover the true data distribution with a modest number of reverse models that can be trained fully parallel, and without weight-sharing, we model each reverse process autoregressively. We use an encoder-decoder transformer architecture [75], such that the decoder models the reverse process for $x_{t-1}$ autoregressively with the help of global context obtained by cross-attending to the encoder's representation of $x_t$ as visualized in Fig. 1. Note that the need for autoregressive modeling gets reduced for small $\beta_t$, which we can adjust for by reducing the number of decoder layers compared to encoder layers. The use of the compression model described in Sec. 3.2, however, allows to utilize full-attention based transformer architectures to implement the autoregressive scales.

## 4  Experiments

Sec. 4.1 evaluates the quality ImageBART achieves in image synthesis. Since we especially want to increase the controllability of the generative process, we evaluate the performance of ImageBART

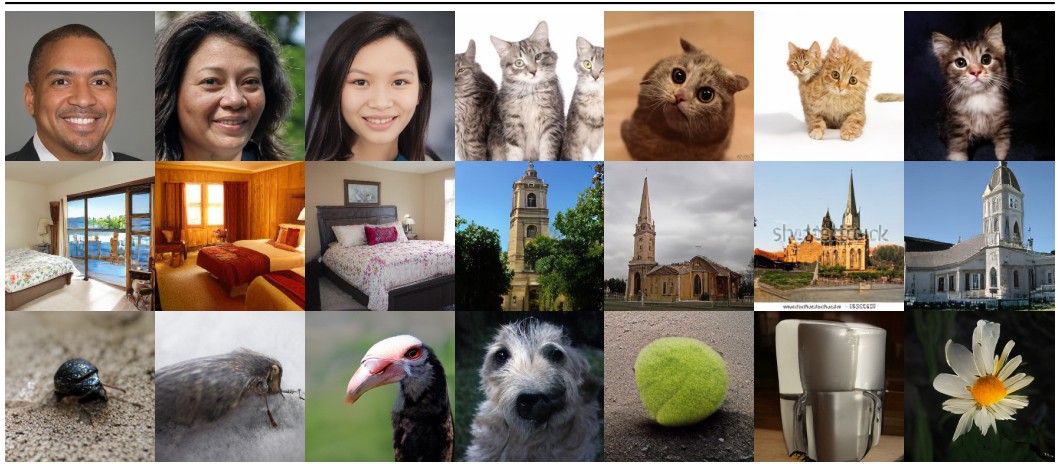

Figure 2: Samples from our models. Top row: FFHQ, LSUN-Cats, Middle row: LSUN-Bedrooms, LSUN-Churches, Bottom row: ImageNet.

| Method | Cats | Beds | Churches | FFHQ |
|---|---|---|---|---|
| VDVAE [10] | – | – | – | 28.5 |
| DDPM [29] | 19.75 | 4.90 | 7.89 | – |
| StyleGAN2 [34] | 7.25 | 2.35 | 3.86 | 3.8 |
| BigGAN [6] | – | – | – | 12.4 |
| DCT [50] | – | 6.40 | 7.56 | 13.06 |
| TT [21] | 17.31 | 6.35 | 7.81 | 11.4 |
| ImageBART | 15.09 | 5.51 | 7.32 | 9.57 |

|  | ImageBART | DDPM | SSDE |
|---|---|---|---|
| *Churches* | | | |
| *Cats* | | | |
| *cIN (c14)* | | | |
| *cIN (c323)* | | | |
| *cIN (c963)* | | | |

Table 1: *Left:* FIDs on the LSUN-{Churches,Beds,Cats} [79] and FFHQ [33] datasets. *Right:* Corresponding qualitative comparisons. Qualitative comparisons with TT can be found in Fig. 20 and Fig. 21

on class- and text-conditional image generation in Sec. 4.2. The ability of our approach to attend to global context enables a new level of localized control which is not possible with previous, purely autoregressive approaches as demonstrated in Sec. 4.3. Finally, Sec. 4.4 presents ablations on model and architecture choices.

### 4.1 High-Fidelity Image Synthesis with ImageBART

In this section we present qualitative and quantitative results on images synthesized by our approach. We train models at resolution $256 \times 256$ for unconditional generation on FFHQ [33], LSUN -Cats, -Churches and -Bedrooms [79] and on class-conditional synthesis on ImageNet (cIN) [13].

**Effective Discrete Representations** Learning the full hierarchy as described in Eq. (2) and without unnecessary redundancies in the data requires to first learn a strong compression model via the objective in Eq. (4). [21] demonstrated how to effectively train such a model and we directly utilize the publicly available pretrained models. For training on LSUN, we finetune an ImageNet pretrained model for one epoch on each dataset. As the majority of codebook entries remains unused, we shrink the codebook to those entries which are actually used (evaluated on the validation split of ImageNet) and assign a random entry for eventual outliers. This procedure yields an effective, compact representation on which we subsequently train ImageBART.

**Training Details** As described in Sec. 3.3, we use an encoder-decoder structure to model the reverse Markov Chain $p_\theta^{t-1}(x_{t-1}|x_t)$, $t < T$, where the encoder is a bidirectional transformer model and decoder is implemented as an AR transformer. As the context for the last scale is pure noise, we employ a decoder-only variant to model $p_\theta^{T-1}(x_{T-1}|x_T)$. Furthermore, to account for the different complexities of the datasets, we adjust the number of multinomial diffusion steps for each dataset accordingly. For FFHQ we choose a chain of length $T = 3$, such that the total model consists of (i) the compression stage and (ii) $n = 2$ transformer models trained in parallel via the objective described in Eq.(7). Similarly, we set $n = 3$ for each of the LSUN models and $n = 5$ for the ImageNet model.

| | rejection rate for cIN sampling | | | | | Text-conditional image synthesis on CC [64] | | | |
|---|---|---|---|---|---|---|---|---|---|
| | 1.0 | 0.5 | 0.25 | 0.05 | | Method | FID $\downarrow$ | IS $\uparrow$ | CLIP-score $\uparrow$ |
| FID | 21.19 | 13.12 | 9.77 | 7.44 | | TT [21] | 28.86 | $13.11_{\pm0.43}$ | $0.20_{\pm0.03}$ |
| IS | $61.6_{\pm0.8}$ | $109.5_{\pm2.3}$ | $146.2_{\pm3.8}$ | $273.5_{\pm4.1}$ | | ImageBART | 22.61 | $15.27_{\pm0.59}$ | $0.23_{\pm0.03}$ |

Table 2: Quantitative analysis on conditional models. Left: Results on class conditional Imagenet for different rejection rates, see also Fig, 20 in the supplemental. Right: Results of text-conditional ImageBART and comparison with TT [21] on the CC test set. Corresponding qualitative comparisons can be found in Fig. 21.

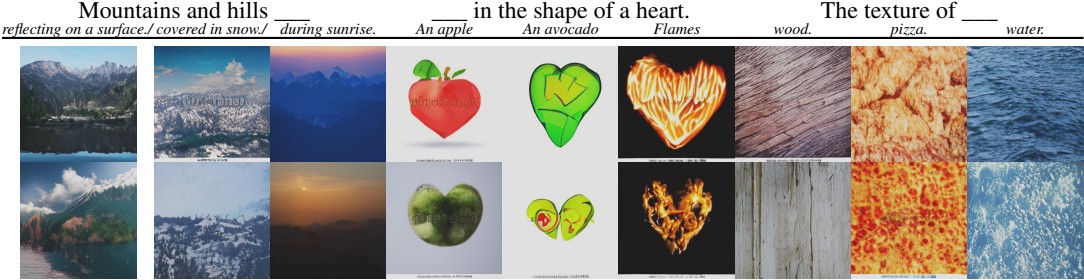

Figure 3: Samples from text-conditional ImageBART. Best 2 of 32 with reranking as in [56].

**Results** For each of these settings, Fig. 2 depicts samples of size $256 \times 256$ generated with Image-BART and a single pass through the learned Markov Chain, demonstrating that our model is able to produce realistic and coherent samples. This is further confirmed by a quantitative analysis in Tab. 1, where we compare FID scores of competing likelihood-based and score-based methods such as TT [21] and DDPM [29]. Regarding other works on diffusion models such as [29] and [68] operating directly in pixel space, we observe that these approaches perform roughly equivalently well in terms of FID for datasets of low complexity (e.g. LSUN-Bedrooms and-Churches). For more complex datasets (LSUN-Cats, cIN), however, our method outperforms these pixel-based approaches, which can also be seen qualitatively on the right in Tab. 1. See Fig. 20 for a comparison on ImageNet.

## 4.2 Conditional Markov Chains for Controlled Image Synthesis

Being a sequence-to-sequence model, our approach allows for flexible and arbitrary conditioning by simply preprending tokens, similar to [21, 56]. More specifically, each learned transition $p_\theta^{t-1}(x_{t-1}|x_t, c)$, $t > 1$ of the Markov chain is then additionally conditioned on a representation $c$, e.g. a single token in the case of the class-conditional model of Sec. 4.1. Note that the compression model $p_\theta^0$ remains unchanged.

**Text-to-Image Synthesis** Besides class-conditional modeling on ImageNet, we also learn a text-conditional model on *Conceptual Captions* (CC) [64, 51]. We obtain $c$ by using the publicly available tokenizer of the CLIP model [55], yielding a conditioning sequence of length 77. To model the dataset, we choose $T = 5$ and thus train $n = 4$ transformer models independently. For the $p_\theta^0$, we directly transfer the compression model from Sec. 4.1, trained on the ImageNet dataset.

Fig. 3 visualizes synthetic samples obtained with this model for various "image-cloze" tasks. Our resulting model is able to attend to semantic variations in the conditioning sentence (e.g. a change of weather for imagery of mountains) and renders the corresponding images accordingly. In Tab. 2, we evaluate FID [28] and Inception Scores (IS) [61] to measure the quality of synthesized images, as well as cosine similarity between CLIP [55] embeddings of the text prompts and the synthesized images to measure how well the image reflects the text. ImageBART improves all metrics upon [21]. Fig. 21 in the supplement provides corresponding qualitative examples for user-defined text inputs.

**Resolutions Beyond 256×256 Pixels.** Our approach is not restricted to generating images of size $256 \times 256$ pixels. Although trained on a fixed resolution, we can apply our models in a patch-wise manner, where we use the sliding attention window of [21] for each scale $t > 0$. As we now incorporate more and more global context while decoding with the Markov chain (which can be thought of as widening a noisy receptive field), ImageBART is able to render consistent images in the megapixel regime. See for example Fig. 4, where we use our text-conditional model to render an

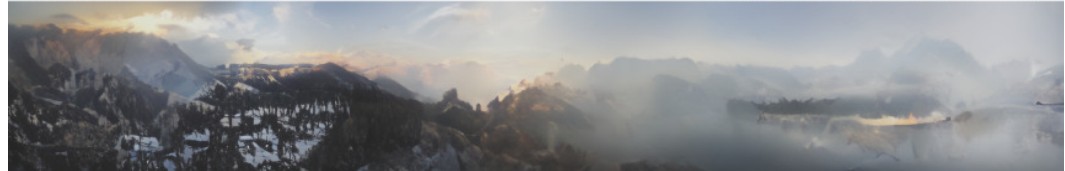

Figure 4: ImageBART is capable of generating high-resolution images. Here, we condition it on text prompts and interpolate between the two descriptions depicted above the image (see also Sec. 4.2).

image of size $300 \times 1800$ pixel and interpolate between two different text prompts. More examples, especially also for semantically guided synthesis, can be found in Sec. A.2.

### 4.3 Beyond Conditional Models: Local Editing with Autoregressive Models

Recent autoregressive approaches, which use a CNN to learn a discrete representation [74], partially alleviate the issues of pixel-wise autoregressive models by working on larger image patches. However, as we show in Fig. 5, even approaches which use adversarial learning to maximize the amount of context encoded in the discrete representation [21] cannot produce completions of the upper half of an image which are consistent with a given lower half.

While our approach also models each transition autoregressively from the top-left to the bottom-right, the ability to attend to global context from the previous scale enables consistent completions of arbitrary order, e.g. right-to-left. To achieve this, we mask the diffusion process as described in Sec. A.3. For a user-specified mask $m$ (e.g. the upper half of an image as in Fig. 5), this results in a forward-backward process $p_\theta^{t-1|t-1,m}$, which, by definition, leaves the unmasked context intact. The reverse process then denoises the unmasked entries to make them consistent with the given context.

Fig. 5 (bottom) visualizes this mixing process, where we use a model with $T = 3$. The first column shows the masked input. To start the process we set all masked entries to random entries. The first two columns then show (decoded) samples from the masked reverse processes $p_\theta^{2,m}$ and $p_\theta^{1,m}$, which still display inconsistencies. The remaining columns show the trajectory of the process $p_\theta^{1|1,m}$, which demonstrates how the model iteratively adjusts its samples according to the given context until it converges to a globally consistent sample. For illustration, we show the analog trajectory obtained with [21], but because it can only attend to unidirectional context, this trajectory is equivalent to a sequence of independent samples and therefore fails to achieve global consistency.

The masked process can be used with arbitrary masks, which enables localized image editing with free, hand-drawn masks as shown in Fig. 6. Note that our model does not need to be trained specifically for this task, which also avoids generalization problems associated with training on masks [81]. Combining this property with the conditional models from Sec. 4.2 allows for especially interesting novel applications, where local image regions are modified based on user specified class or text prompts, as shown in Fig. 7.

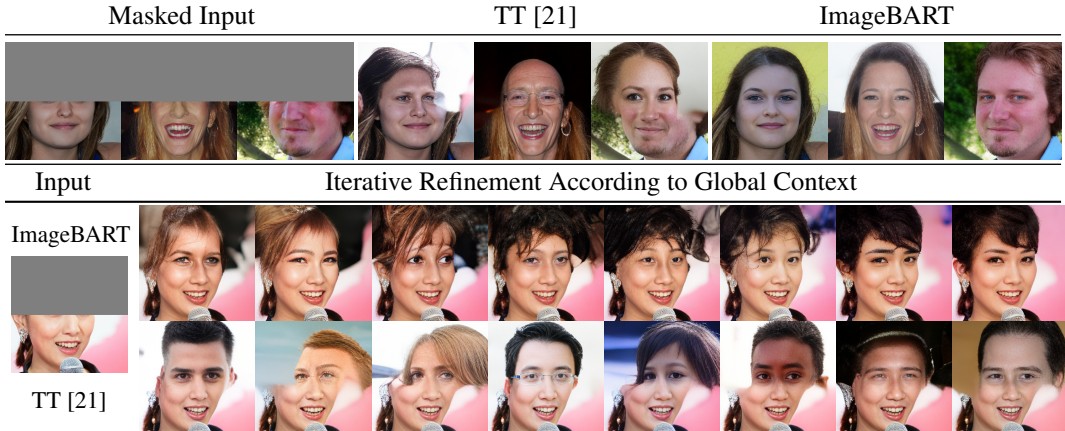

Figure 5: Without global context, AR models fail at completing upper halfs, contrasting ImageBART.

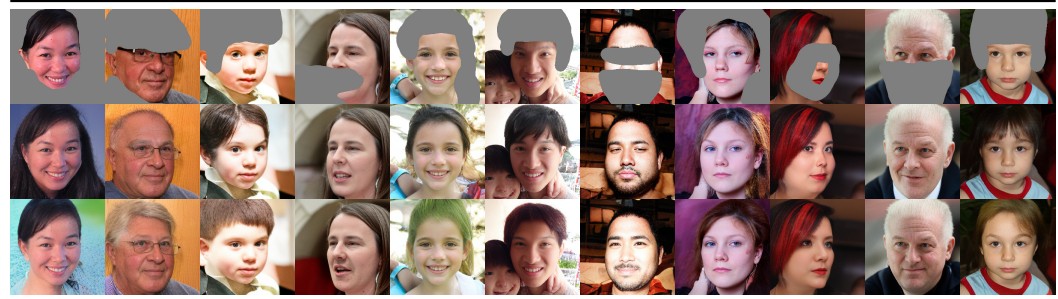

Figure 6: Local editing application using markov chain of length 16 on FFHQ. By incorporating bidirectional context ImageBART is able to solve this unconditional inpainting task (cf. Sec. 4.3).

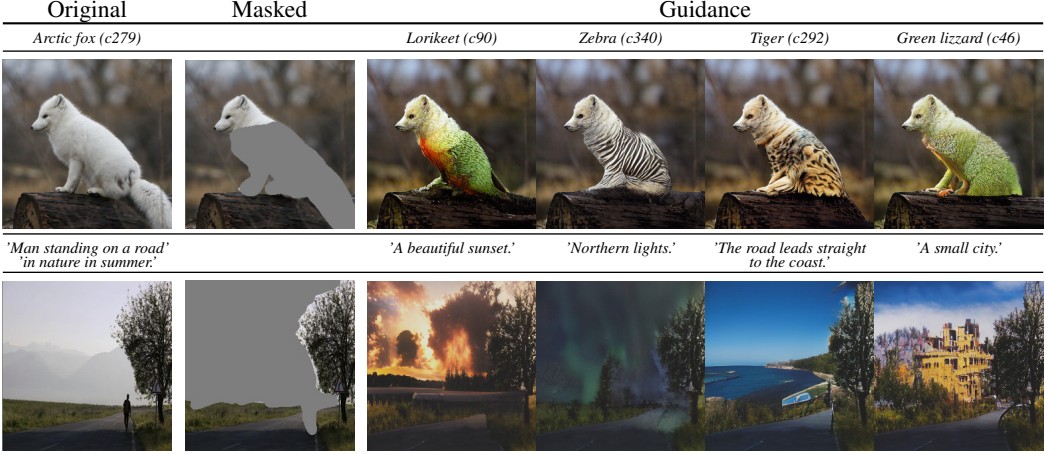

Figure 7: Conditionally guided inpainting results obtained from conditional ImageBART trained on the i) ImageNet (top row) and ii) Conceptual Captions (bottom row) datasets.

## 4.4 Ablations

**On the Number of Diffusion Steps** In this section we analyze the effect of varying the number of diffusion steps (denoted by $T$). To do so, we perform an experiment for unconditional training on the FFHQ dataset, where we train a Taming Transformers (TT) baseline (corresponding to the case $T = 2$ within our framework) with 800M parameters and three variants of ImageBART with $T = 3$ (2x400M), $T = 5$ (4x200M) and $T = 9$ (8x100M), respectively. Note that for a fair comparison, all models use the same first level for compression, and we fix the number of remaining parameters to 800M and distribute them equally across all scales. All models were trained with the same computational budget and evaluated at the best validation checkpoint.

In Tab. 3, we assess both the pure synthesis and the modification ability of ImageBART by computing FID scores on samples and modified images (in the case of upper half completion as in Fig. 5). For both tasks, we use a single pass through the reverse Markov chain. We observe that the modification performance increases monotonically with the number of scales, which highlights the improved image manipulation abilities of our approach. For unconditional generation, we observe a similar trend, although FID seems to plateau beyond $T = 5$.

**Joint vs. Independent Training** While it is possible to optimize Eq. (2) jointly across all scales, we found that training is more robust when training all scales independently. Besides the usual separation of training the compression model $p_\theta^0$ and the generative model $p_\theta^{t \geq 1}$, training the latter in parallel over multiple scales avoids the tedious weighting of the loss contribution from different scales; an often encountered problem in other denoising diffusion probabilistic models [29].

**Efficiency with Less Decoder Layers** As we implement the conditional transition probabilities $p_\theta^{t-1}$ with an encoder-decoder transformer architecture, we are interested in the effect of altering the ratio of encoder and decoder layers in the model. Recent work has provided evidence that it is possible to

| Unconditional Generation | | | Upper Half Completion | | |
| --- | --- | --- | --- | --- | --- |
| method | FID ↓ | IS ↑ | method | FID ↓ | IS ↑ |
| TT ($T = 2$) | 12.44 | $4.42 \pm 0.05$ | TT ($T = 2$) | 11.80 | $4.48 \pm 0.10$ |
| ImageBART ($T = 3$) | 12.55 | $3.98 \pm 0.07$ | ImageBART ($T = 3$) | 9.25 | $4.49 \pm 0.13$ |
| ImageBART ($T = 5$) | 10.69 | $4.27 \pm 0.05$ | ImageBART ($T = 5$) | 6.87 | $4.81 \pm 0.13$ |
| ImageBART ($T = 9$) | 10.81 | $4.49 \pm 0.05$ | ImageBART ($T = 9$) | 6.64 | $4.86 \pm 0.15$ |

Table 3: Assessing the effect of different $T$ with a fixed number of parameters distributed equally over all scales. All models are trained on FFHQ. *Left:* Full image generation results. *Right:* Using the example of upper image completion, we evaluate the ability to complete and modifiy an image, see Sec. 4.3 and 4.4.

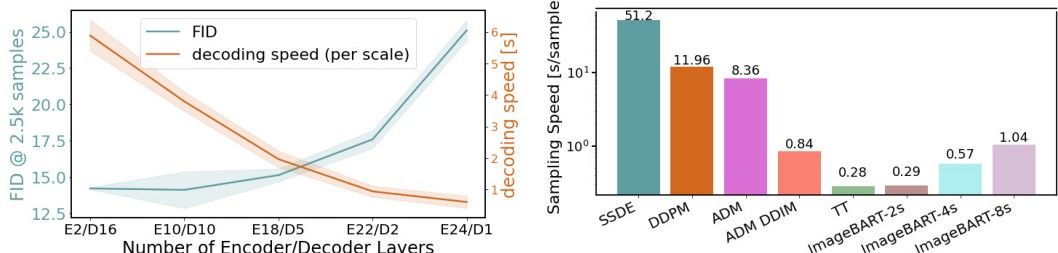

Figure 8: *Left:* Effect of number of encoder vs. decoder layers for a fixed total number of model parameters $((195 \pm 5)M)$, evaluated on LSUN-Churches. FIDs are evaluated w.r.t $3 \times 2500k$ samples. The plot shows 3 standard deviations. All models are trained jointly over three scales. *Right:* Our model achieves better sampling performance than state of the art diffusion models (SSDE [68], DDPM [29], ADM [15]) and also approaches the inference speed of TT [21], which only consists of a single autoregressive stage. Reducing the number of scales increases inference speed at the expense of controllability. Experiments were conducted on a single NVIDIA A100 and are reported averaged over 1000 samples with a batch size of 50, evaluated on FFHQ while using the same number of trainable parameters (800m) for all AR models.

significantly reduce the number of decoder layers and thus also decrease autoregressive decoding speed while maintaining high quality [35]. We perform an experiment on LSUN-Churches, where we analyze the effect of different layer-ratios on synthesis quality (measured by FID) and on decoding speed when fixing the total number of model parameters to 200M. The results in the left part of Fig. 8 confirms that it is indeed possible to reduce the number of decoder layers while maintaining satisfactory FID scores with higher decoding efficiency. We identity a favorable trade-off between four and six decoder layers and transfer this setting to our other experiments.

Finally, we compare our model in terms of sampling speed with the recent state-of-the-art generative diffusion [29, 68] and AR models [21]. The results are summarized in Fig. 8. While consistently being faster than all pixel-based models due to training in a compressed latent space, the increase in runtime w.r.t. [21] is moderate due to the use of encoder-decoder transformers, i.e., a a decrease in pure decoder layers. If a faster runtime is desired, the speed can be further increased by reducing the number of decoder layers even more, see also the discussion in Sec. A.5.

## 5   Conclusion

We have proposed ImageBART, a hierarchical approach to introduce bidirectional context into autoregressive transformer models for high-fidelity controllable image synthesis. We invert a multinomial diffusion process by training a Markov chain to gradually incorporate context in a coarse-to-fine manner. Our study shows that this approach (i) introduces a natural hierarchical representation of images, with consecutive levels carrying more information than previous ones. (see also Fig. 9). (ii) It alleviates the unnatural unidirectional ordering of pure autoregressive models for image representation through global context from previous levels of the hierarchy. (iii) It enables global and local manipulation of a given input, a feat previously out-of-reach for ARMs. (iv) We additionally show that our model can be efficiently conditioned on various representations, allowing for a large class of conditional image synthesis tasks such as semantically guided generation or text-to-image synthesis.

## Acknowledgments

Many thanks to Phil Wang for providing `https://github.com/lucidrains/x-transformers` and all the other great PyTorch implementations.

## Funding and Transparency Statement

Funding in direct support of this work: German Research Foundation (DFG) projects 371923335 and 421703927, German Federal Ministry for Economic Affairs and Energy within the project 'KI-Absicherung - Safe AI for automated driving'.

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
