# A  Appendix

## A.1  Hyperparameters & Implementation Details

### A.1.1  Compression Models

| experiment | section | add. samples | effective size | fine-tuned from [21] | trained from scratch | compression rate |
|---|---|---|---|---|---|---|
| *class-cond. ImageNet* | 4.1 | Fig. 10,18,19 | 973 | ✗ | ✗ | $^1/_{256}$ |
| *LSUN-Cats* | 4.1 | Fig. 23 | 1014 | ✓ | ✗ | $^1/_{256}$ |
| *LSUN-Churches* | 4.1 | Fig. 22 | 1022 | ✓ | ✗ | $^1/_{256}$ |
| *LSUN-Bedrooms* | 4.1 | Fig. 24 | 1017 | ✓ | ✗ | $^1/_{256}$ |
| *Conceptual Captions* | 4.2 | Fig. 11 | 973 | ✗ | ✗ | $^1/_{256}$ |
| *FFHQ* | 4.1 | Fig. 25 | 548 | ✗ | ✓ | $^1/_{256}$ |
| *Semantic FLICKR* | A.2 | Fig. 12,13 | 973 | ✗ | ✗ | $^1/_{256}$ |

Table 4:  Hyperparameters for all compression models used in our experiments.

We follow [21] and implement our image compression models as "VQGANs".  More specifically, we use the official implementation provided at `https://github.com/CompVis/taming-transformers` and fine-tune the publicly available model for experiments on LSUN. For FFHQ, we train such a compression model from scratch. See Tab. 4 for an overview. As some of the codebook entries remain unused after training, we shrink the codebook to its *effective* size when training a generative model on top of it. For eventual entries not detected during evaluation on the subset, we assign a random entry.

### A.1.2  Hierarchical Representations via Multinomial Diffusion

Tab. 5 lists the configurations of the multinomial diffusion processes for each experiment described in this work (see also Tab. 4). Note that all representations $x_t$ for $T > 1$ have the same spatial resolution, but since each forward diffusion process gradually removes information, we obtain a coarse-to-fine hierarchy. On average, level $x_t$ will contain $\lfloor \bar{\alpha}_t \cdot N \rfloor$ valid entries, which we denote as the effective sequence length in Tab. 5. Thus, ImageBART can also be interpreted as a generative compression model as illustrated in Fig. 9: By trading perfect reconstruction quality for compression, one can obtain a significantly shorter sequence, still representing a visually plausible image. This provides the basis for learning a generative model that does not waste capacity on redundancies in the data [17] and the compressed space significantly lowers the computational demands for training and decoding.

### A.1.3  Reverse Diffusion with Transformer Models

ImageBART is a learned Markov chain, trained to reverse the multinomial diffusion process described in Eq. (5). We can efficiently model the conditionals $p_\theta^t$ with a sequence-to-sequence model and follow [75, 45] to implement $p_\theta^t$ with an encoder-decoder architecture.  Tab. 6 summarizes the hyperparameters used to implement the conditionals for each experiment.  For comparison, the

| experiment | length of chain | $\beta_t$ schedule ($t \geq 2$) | effect. seq. length (full $N = 256$, $t \geq 2$, w/o cond.) |
|---|---|---|---|
| *class-cond. ImageNet* | $T = 6$ | $[0.090, 0.104, 0.139, 0.266, 1.0]$ | $[232, 208, 179, 131, 0]$ |
| *LSUN-Cats* | $T = 4$ | $[0.152, 0.231, 1.0]$ | $[217, 166, 0]$ |
| *LSUN-Churches* | $T = 4$ | $[0.152, 0.231, 1.0]$ | $[217, 166, 0]$ |
| *LSUN-Bedrooms* | $T = 4$ | $[0.152, 0.231, 1.0]$ | $[217, 166, 0]$ |
| *Conceptual Captions* | $T = 5$ | $[0.113, 0.141, 0.246, 1.0]$ | $[227, 195, 147, 0]$ |
| *FFHQ* | $T = 3$ | $[0.364, 1.0]$ | $[162, 0]$ |
| *Semantic FLICKR* | $T = 5$ | $[0.250, 0.333, 0.500, 1.0]$ | $[192, 128, 64, 0]$ |

Table 5:  Hyperparameters for all multinomial diffusion process we used in our experiments.

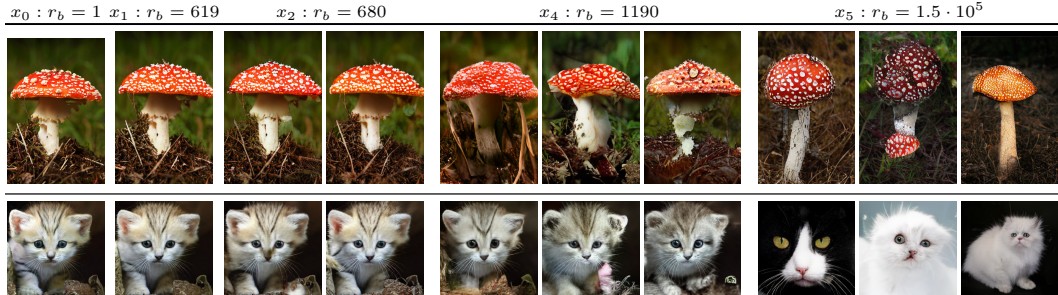

$x_0 : r_b = 1$  $x_1 : r_b = 619$  $x_2 : r_b = 680$  $x_4 : r_b = 1190$  $x_5 : r_b = 1.5 \cdot 10^5$

Figure 9: ImageBART can be interpreted as a generative compression model retaining high visual quality at high compression rates. Here, we denote the compression rate $r_b = \frac{256 \cdot 256 \cdot 3 \cdot 8}{n_{eff} \cdot \log_2(973)}$ (in bits), and $n_{eff}$ is the effective sequence length as in Tab. 5.

| experiment | num. parameters [M] | num. layers (encoder/decoder) | embed. dim. (encoder & decoder) |
|---|---|---|---|
| *class-cond. ImageNet* | $[693, 693, 693, 693, 718]$ | $[4 \times (32/6), (0/36)]$ | $[4 \times 1152, 1216]$ |
| *LSUN-Cats* | $[693, 693, 718]$ | $[2 \times (32/6), (0/36)]$ | $[2 \times 1152, 1216]$ |
| *LSUN-Churches* | $[693, 693, 718]$ | $[2 \times (32/6), (0/36)]$ | $[2 \times 1152, 1216]$ |
| *LSUN-Bedrooms* | $[693, 693, 718]$ | $[2 \times (32/6), (0/36)]$ | $[2 \times 1152, 1216]$ |
| *Conceptual Captions* | $[685, 685, 685, 778]$ | $[3 \times (32/6), (0/36)]$ | $[3 \times 1152, 1216]$ |
| *FFHQ* | $[687, 713]$ | $[(32/6), (0/36)]$ | $[1152, 1216]$ |
| *Semantic FLICKR* | $[397, 397, 397, 429]$ | $[3 \times (19/5), (0, 24)]$ | $[1152, 1216]$ |

Table 6: Hyperparameters for each experiment and scale ($t \geq 2$) used to implement the Markov chain in Eq. (5).

models in Tab. 1 contain 115M (VDVAE), 255M (DDPM), 30M (StyleGAN2), 158M (BigGAN), 448M (DCT) and 600M (TT) parameters.

### A.1.4   Hardware

All models listed in Tab. 4 and Tab. 6 were optimized on a single NVIDIA A100 GPU and using 32-bit precision. Sampling speed as reported in Fig. 8 was also measured on a NVIDIA A100.

### A.2   Details on Conditional Experiments

**Semantically Guided Synthesis** In addition to class- and text-conditional generative modeling, we apply our model to semantically guided synthesis of landscape images [53]. To achieve this we follow [21] and use the discrete representation of an autoencoder model trained on segmentation masks as conditioning $c$ for our models $p_\theta^t$. However, since simply prepending $c$ here doubles the total length, which means a fourfold increase in complexity in the attention mechanism, we exploit the fact that the segmentation masks and the images (or their representations) are aligned. More specifically, within the encoder-decoder architecture, we first produce two embeddings $e_1$ and $e_2$ for $x_t$ and $c$, respectively, which are subsequently concatenated channel-wise, thereby keeping the sequence length of $x_t$. With this modifications, we train a model with $T = 5$ and individually optimize each scale similar to the unconditional training setting. Here again, we use the compression model $p_\theta^0$ pre-trained on ImageNet. For training, we randomly crop the images and semantic maps to size $256 \times 256$. For testing, however, we again use the sliding window approach of [21] (cf. Sec. 4.2), which enables us to generate high-resolution images of landscapes, as visualized in Fig. 12 and Fig. 13.

### A.3   Masked Diffusion Processes for Local Editing

Previous autoregressive approaches [72] model images directly as a sequence of pixels from the top-left to the bottom-right. Thus, when generating a pixel, only context from neighbors to the left and above can be taken into account. While more recent approaches, which use a CNN to learn a discrete representation that is subsequently modeled autoregressively [74, 21], improve this situation

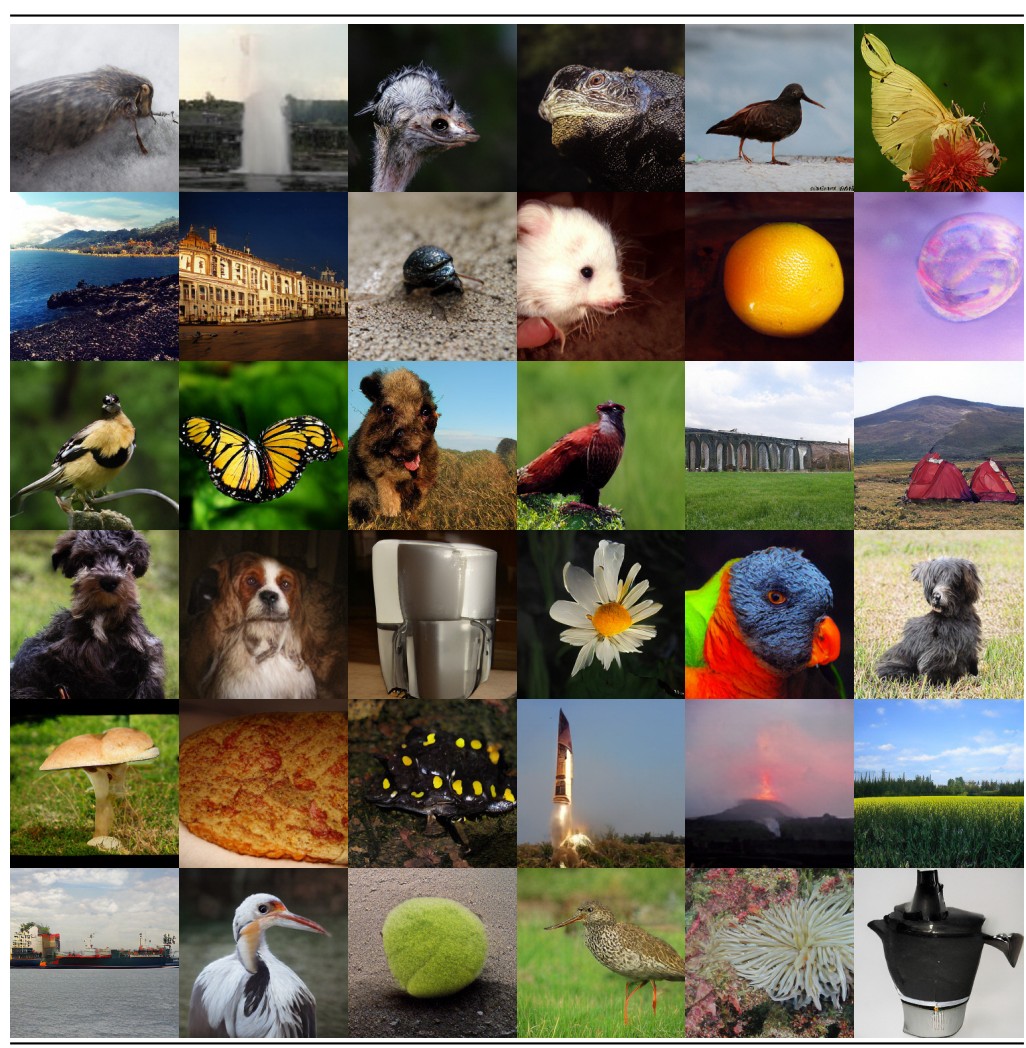

Figure 10: Additional samples for class-conditional synthesis results on ImageNet.

A heart made of ___.                 A vase made of ___.

*wood*      *stone*      *pizza*      *metal*      *wood*      *avocado*      *pizza*      *pasta*

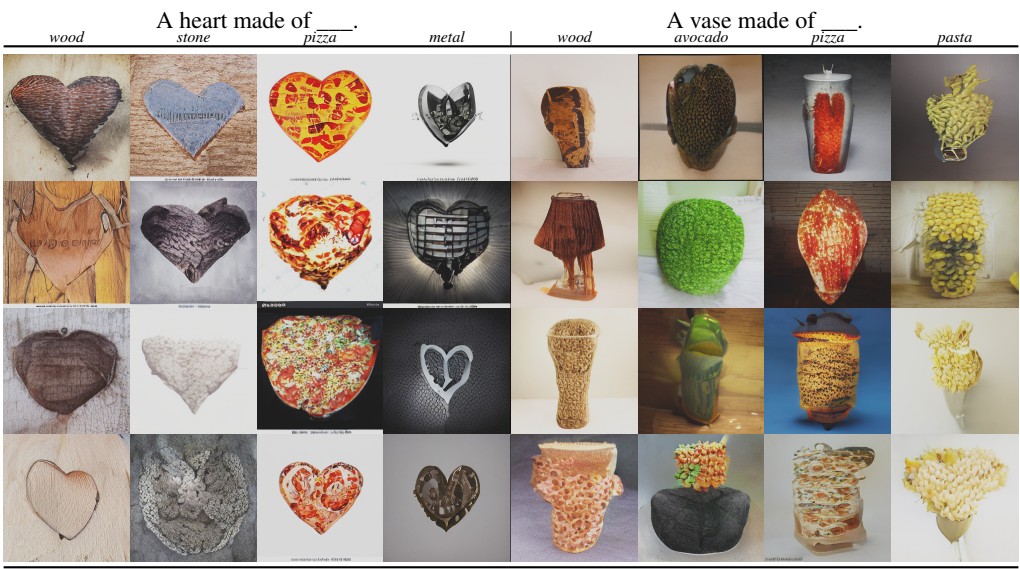

Figure 11: Additional samples from our text-conditional model.

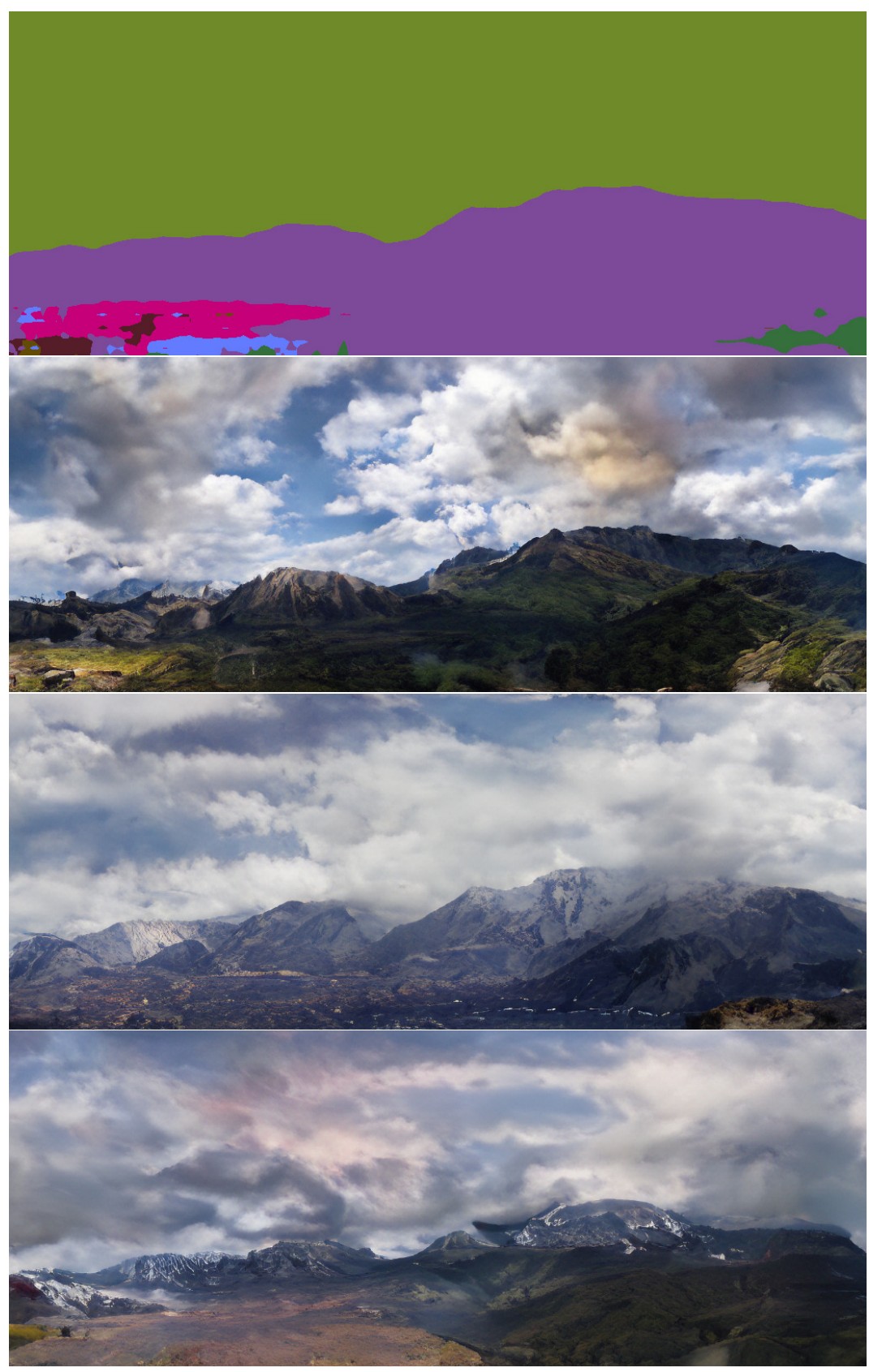

Figure 12: Semantically guided samples from ImageBART conditionally trained semantic maps from the on the S-FLCKR dataset, similar to the one shown in the top row. Image size is $1024 \times 410$.

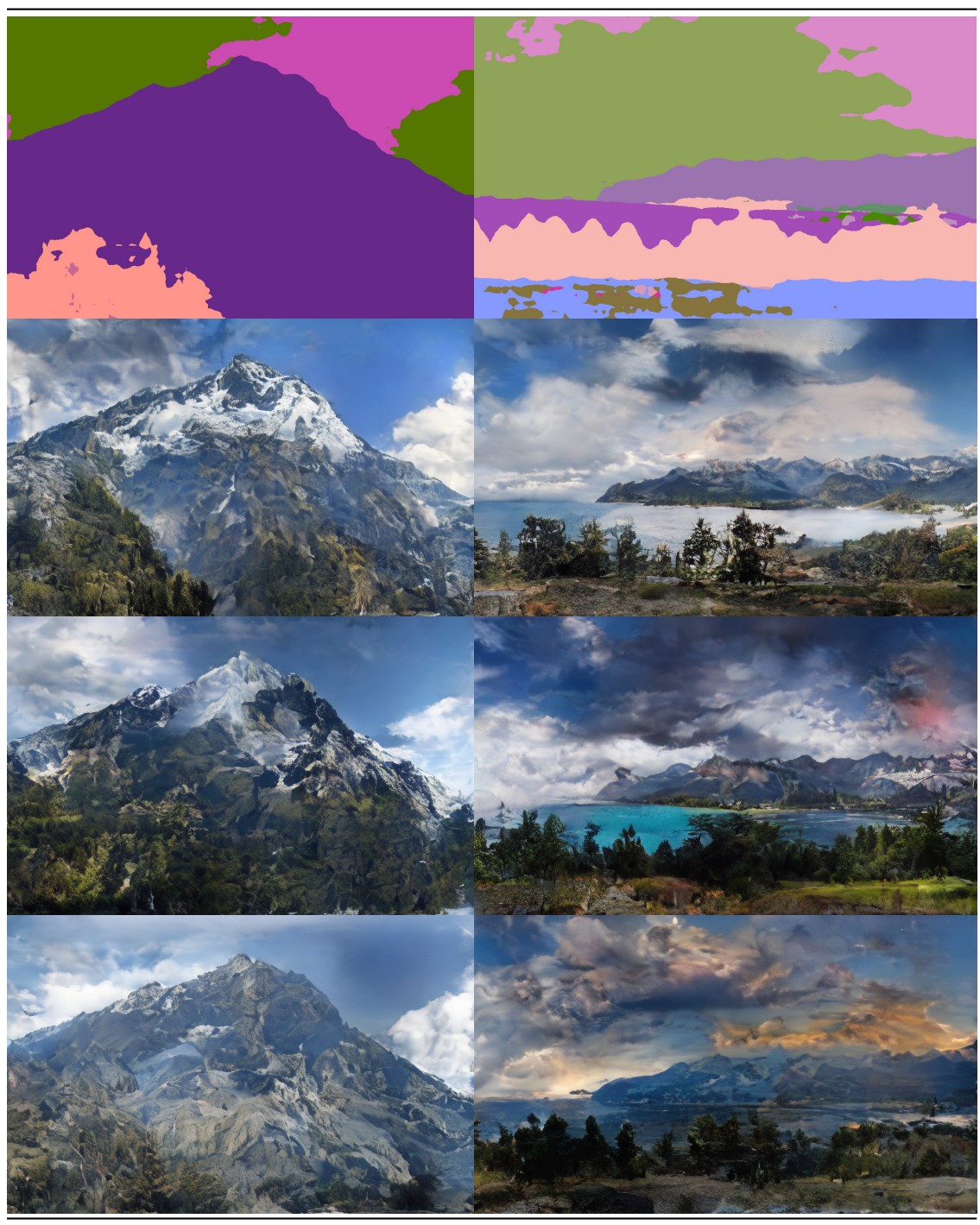

Figure 13: Additional samples on semantic image synthesis. Left: $1024 \times 656$ pix. Right: $1024 \times 608$ pix.

Masked Input            TT [21]            ImageBART

Iterative Refinement According to Global Context ⟶

Input

ImageBART

TT [21]

ImageBART

TT [21]

ImageBART

TT [21]

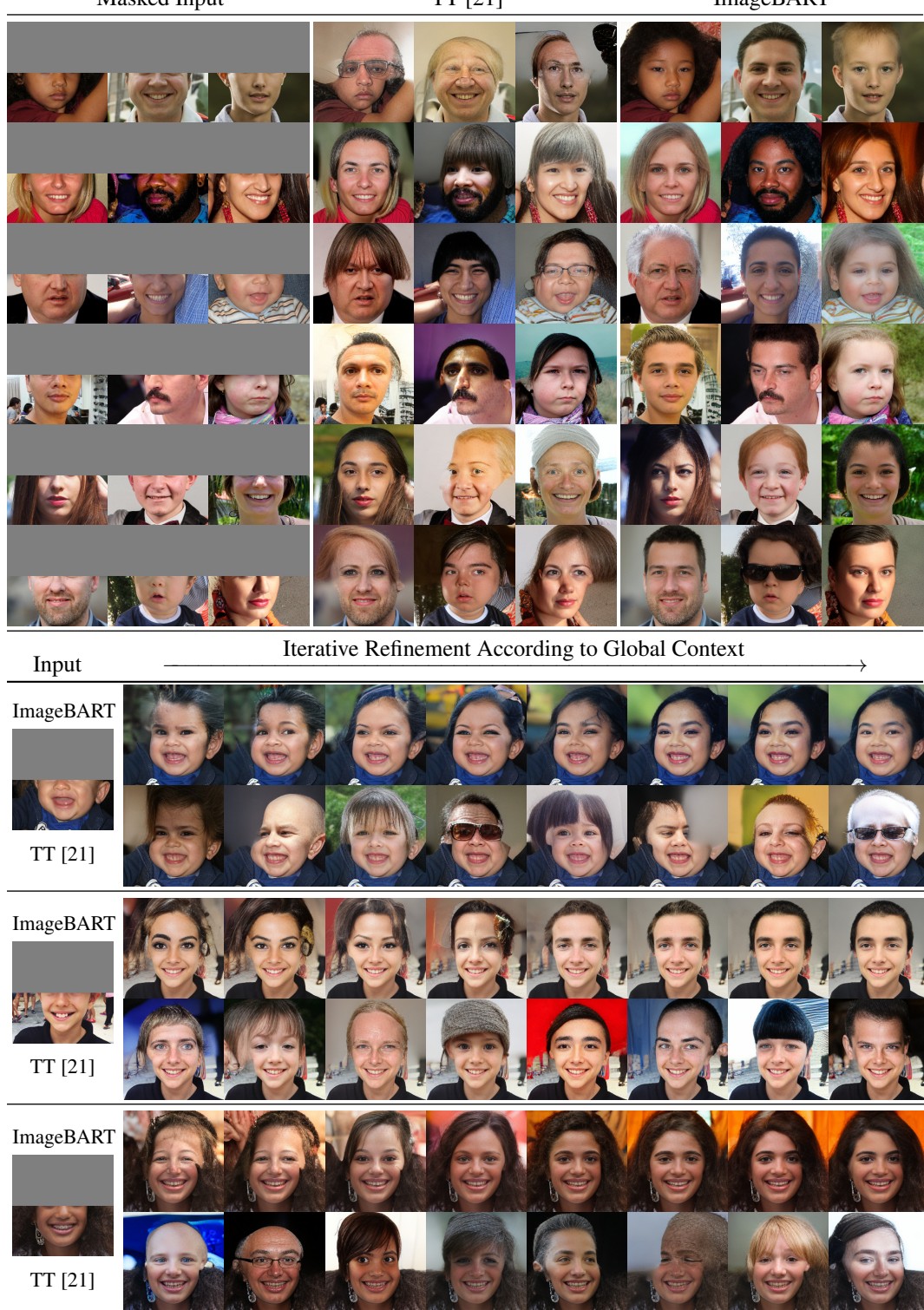

Figure 14: Additional examples for upper half completion as in Fig. 5. The top shows masked inputs, results by TT [21] and results by ImageBART. The bottom shows every other sample of the forward-backward chain described in Sec. 4.3 and Sec. A.3. ImageBART can incorporate global context to produce consistent completions, whereas TT is limited to context from above and thus fails to produce consistent completions.

| Original | Masked | | Guidance | | |
|---|---|---|---|---|---|

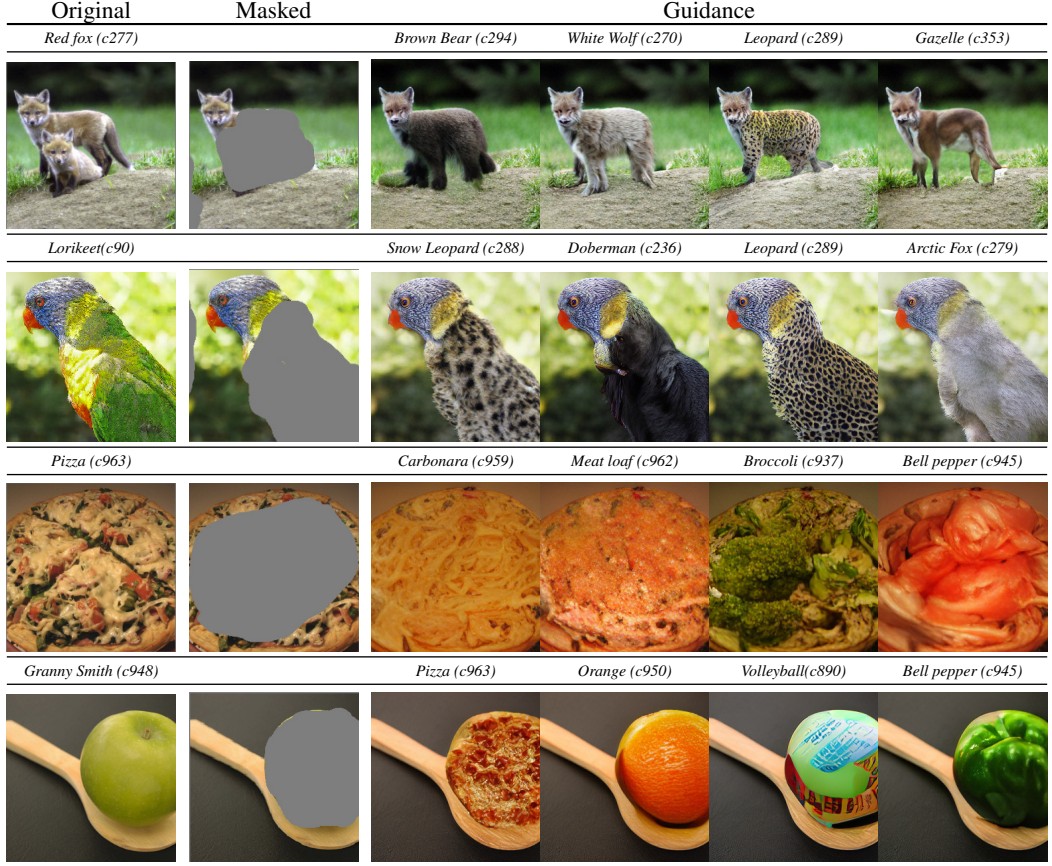

Figure 15: Conditionally guided inpainting results obtained from conditional ImageBART trained on the ImageNet dataset.

| Original | Masked | | Guidance | | |
|---|---|---|---|---|---|

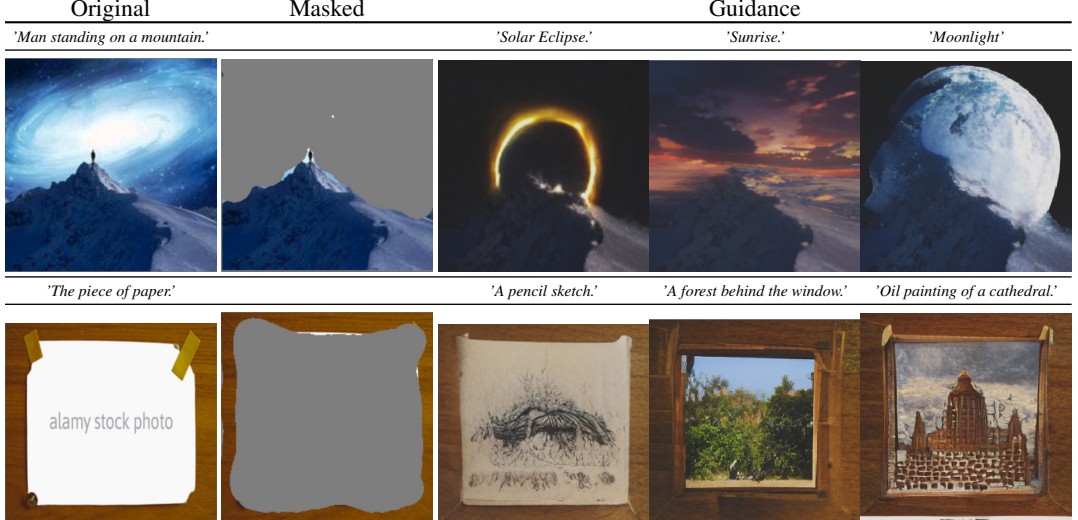

Figure 16: Additional results on conditional inpainting obtained from conditional ImageBART trained on the Conceptual Captions dataset.

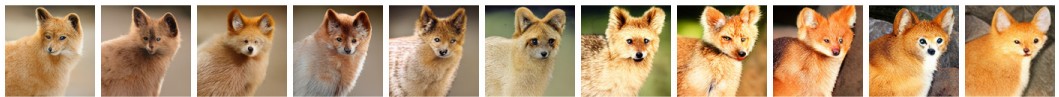

Figure 17: Running the forward-backward chain on a class conditional model allows for fine-grained exploration of samples for a given class, such as red fox (class 277).

because elements of the representation now correspond to image patches, Fig. 5 showed that these models still fail to generate completions of the upper half of an image which are consistent with a given lower half.

While our approach also models each transition autoregressively from the top-left to the bottom-right, each transition additionally has access to global context from the previous step. We aim to exploit this fact to obtain novel applications such as consistent completions of upper halfs and, more generally, completions with respect to an arbitrary mask. For any such mask, let $m$ denote the result of downsampling it to the size of $x_1$ using nearest-neighbor-interpolation, such that $m^i = 0$ gives the positions where context should be used, and $m^i = 1$ gives the positions where new content should be generated. We then define the masked forward process,

$$q_\theta^{t,m}(x_t|x_{t-1}) = m \cdot q_\theta^t(x_t|x_{t-1}) + (1-m) \cdot \delta(x_t - x_{t-1}), \tag{10}$$

which only diffuses masked entries, and the masked reverse process,

$$p_\theta^{t-1,m}(x_{t-1}|x_t) = m \cdot p_\theta^{t-1}(x_{t-1}|x_t) + (1-m) \cdot \delta(x_{t-1} - x_t), \tag{11}$$

which only denoises masked entries. By definition, running this process forward and then backward again represents the identity on umasked entries such that the given context remains constant. We denote this forward-backward process that starts from a given $x_{t-1}$ and produces a sample $x_{t-1,m}$,

$$x_t \sim q_\theta^{t,m}(x_t|x_{t-1}), \quad x_{t-1,m} \sim p_\theta^{t-1,m}(x_{t-1,m}|x_t) \tag{12}$$

by $p_\theta^{t-1|t-1,m}$ and use it to sample with spatial conditioning information. Since it always leaves the unmasked context intact, the reverse process denoises the unmasked entries to make them consistent with the given context.

Besides Fig. 5, 6 additional visualizations of this process can be found in Fig. 14. The top shows masked inputs (left), final results of upper completions obtained by [21] (middle) and by $p_\theta^{1|1,m}$ (right). The bottom visualizes the trajectory of the masked process, showing the masked input (leftmost column), denoised samples from $p_\theta^{2,m}$ (first column) and $p_\theta^{1,m}$ (second column), and every other sample from the forward-backward model $p_\theta^{1|1,m}$. It demonstrates how the model iteratively incorporates global context from the previous scale to converge to a globally consistent sample at the very right. A visualization of the process on the class conditional ImageNet model is shown in Fig. 17. Additional examples for conditional samples from this process, as in Fig. 7, can be found in Fig. 15 and Fig. 16.

## A.4 Limitations and Societal Impacts

Training deep generative models consumes a significant amount of energy (see also Sec. A.1 regarding the used hardware; the ImageNet model for example was trained for 19 days). With regard to the environment, it is important that we reduce the energy consumption as much as possible. To take a step in this direction, we followed previous works and relied on a strongly compressed, learned representation of images. Because we can fine-tune the corresponding encoder and decoder models from pre-trained ones, the costs for this step are largely amortized and subsequent levels of our hierarchy benefit from a drastically reduced sequence length. Nonetheless, it should be noted that such a strong compression scheme for images does not result in perfect reconstructions. For applications which require very high fidelity, such a level of compression might be unsuitable due to artifacts in the reconstructed images. Additionally, the use of adversarial learning in this stage can potentiate biases of datasets by its mode-seeking behavior. Both of these issues can be lessened with larger sequence lengths at the cost of higher energy requirements.

The transformer architecture which is used to model the transitions in our hierarchy is generally considered to be less biased compared to convolutional architectures. However, it also cannot benefit from useful inductive biases and therefore requires a large amount of data and resources to learn all required relationships. In early experiments, we noticed that on small datasets, such as CIFAR-10 [42], the transformer models overfit before they reach good performance. Thus, in its current implementation our approach requires datasets of sufficient size. Future works should evaluate different architectures, regularizations or augmentations to enable its use on small datasets. On the other extreme, we find that with large datasets, the main bottleneck is the computational resources that can be spent on the training of the transformer models. On the largest datasets, *Conceptual Captions* and *ImageNet*, we find that performance still improves after two weeks of training. Thus, consistent with other works on scaling up generative models, we expect that performance of our model will keep increasing with the available resources.

To ensure comparability with other approaches, we use standard benchmark datasets for deep generative models, even if some of them are known to contain offensive content [11].

## A.5 Sampling Speed

Here, we discuss the effects of varying the number of encoder vs. decoder layers in ImageBART on sampling speed as presented in Sec. 4.4. On each diffusion scale, the encoder layers only have to run once whereas the decoder layers have to run $n_{\text{data\_dim}}$ times. This results in an approximate complexity of order $n_{\text{scales}} C(n_{\text{encoder\_layers}} + n_{\text{data\_dim}} n_{\text{decoder\_layers}})$, where $C$ is the complexity of a single transformer layer. The speedup from such an encoder-decoder transformer over a decoder only transformer with $n_{\text{encoder\_layers}} + n_{\text{decoder\_layers}}$ layers is therefore

$$\left( n_{\text{encoder\_layers}} + n_{\text{decoder\_layers}} \right) \cdot \left( \frac{n_{\text{encoder\_layers}}}{n_{\text{data\_dim}}} + n_{\text{decoder\_layers}} \right)^{-1}. \tag{13}$$

## A.6 Additional Samples & Nearest Neighbors

We provide additional samples from our models in Fig. 18-25. Additionally, we also provide nearest neighbors (measured in VGG feature space) for our FFHQ and LSUN-Churches models in Fig. 26 and Fig. 27, respectively.

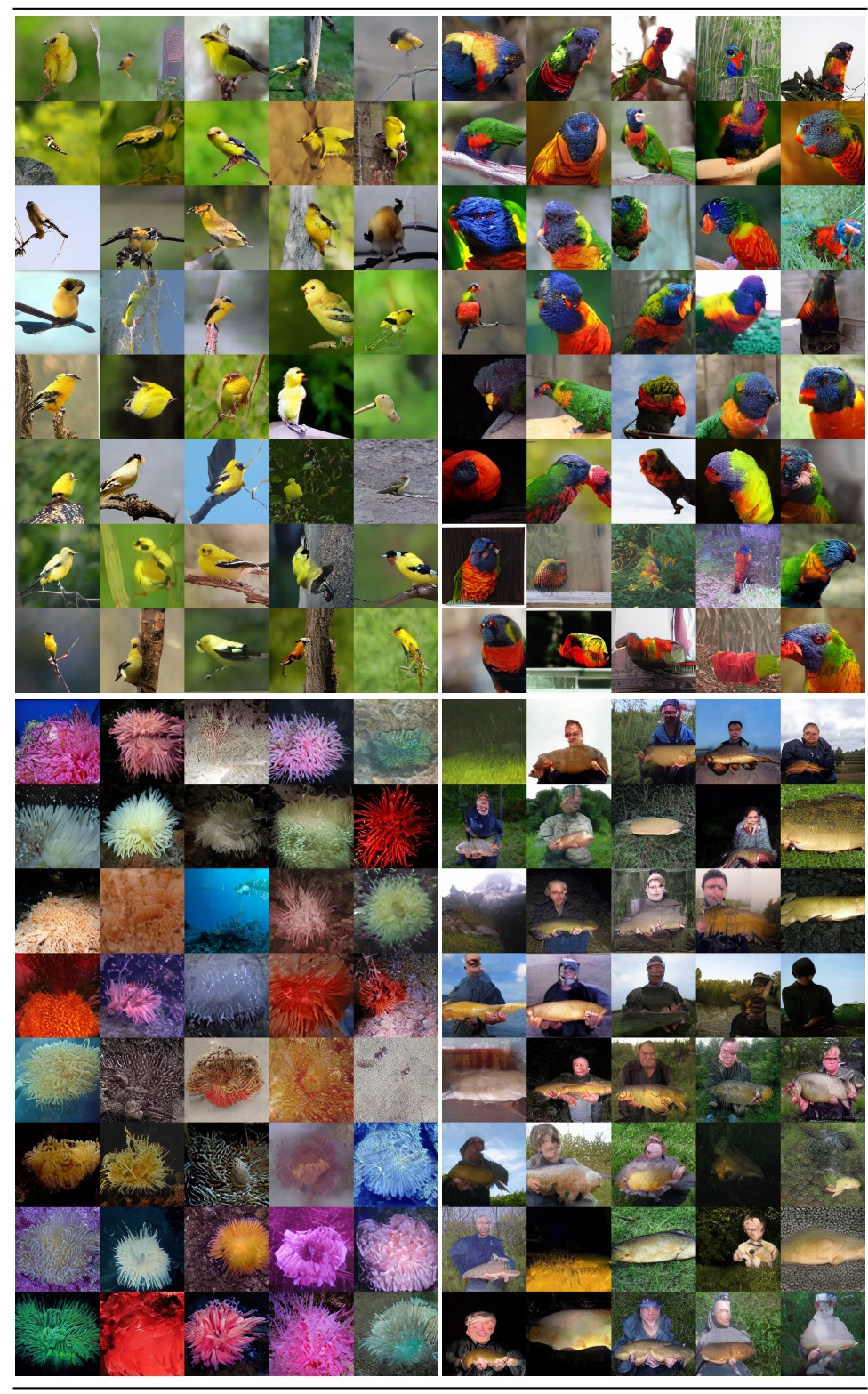

Figure 18: Additional class-conditional 256 × 256 random samples on ImageNet. Depicted classes are *11: goldfinch* (top left), *90: lorikeet* (top right), *108: sea anemone* (bottom left) and *0: tench* (bottom right).

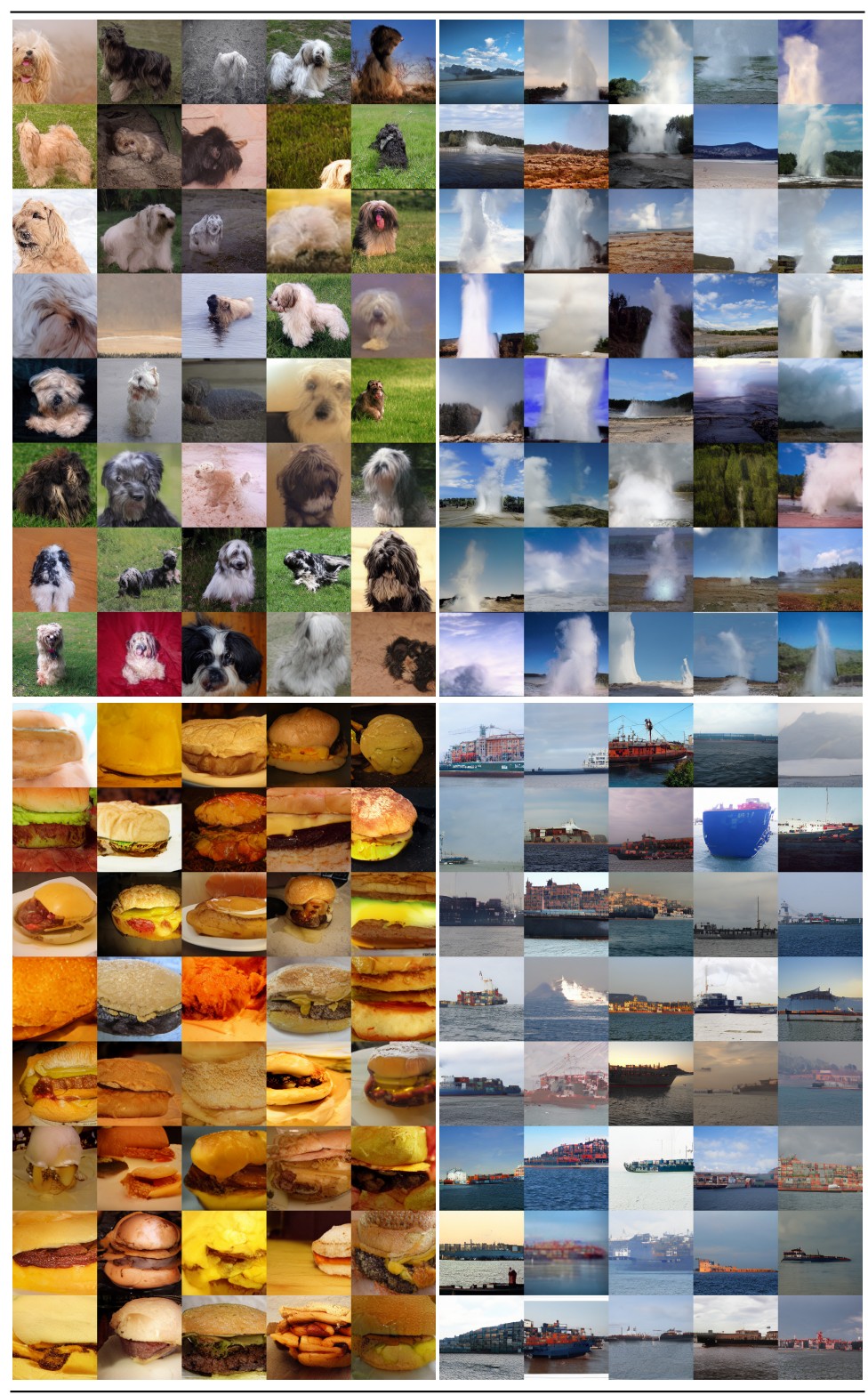

Figure 19: Additional class-conditional $256 \times 256$ random samples on ImageNet. Depicted classes are *200: tibetian terrier* (top left), *974: geyser* (top right), *933: cheeseburger* (bottom left) and *510: container ship* (bottom right).

| rate | ImageBART | TT [21] |
|---|---|---|

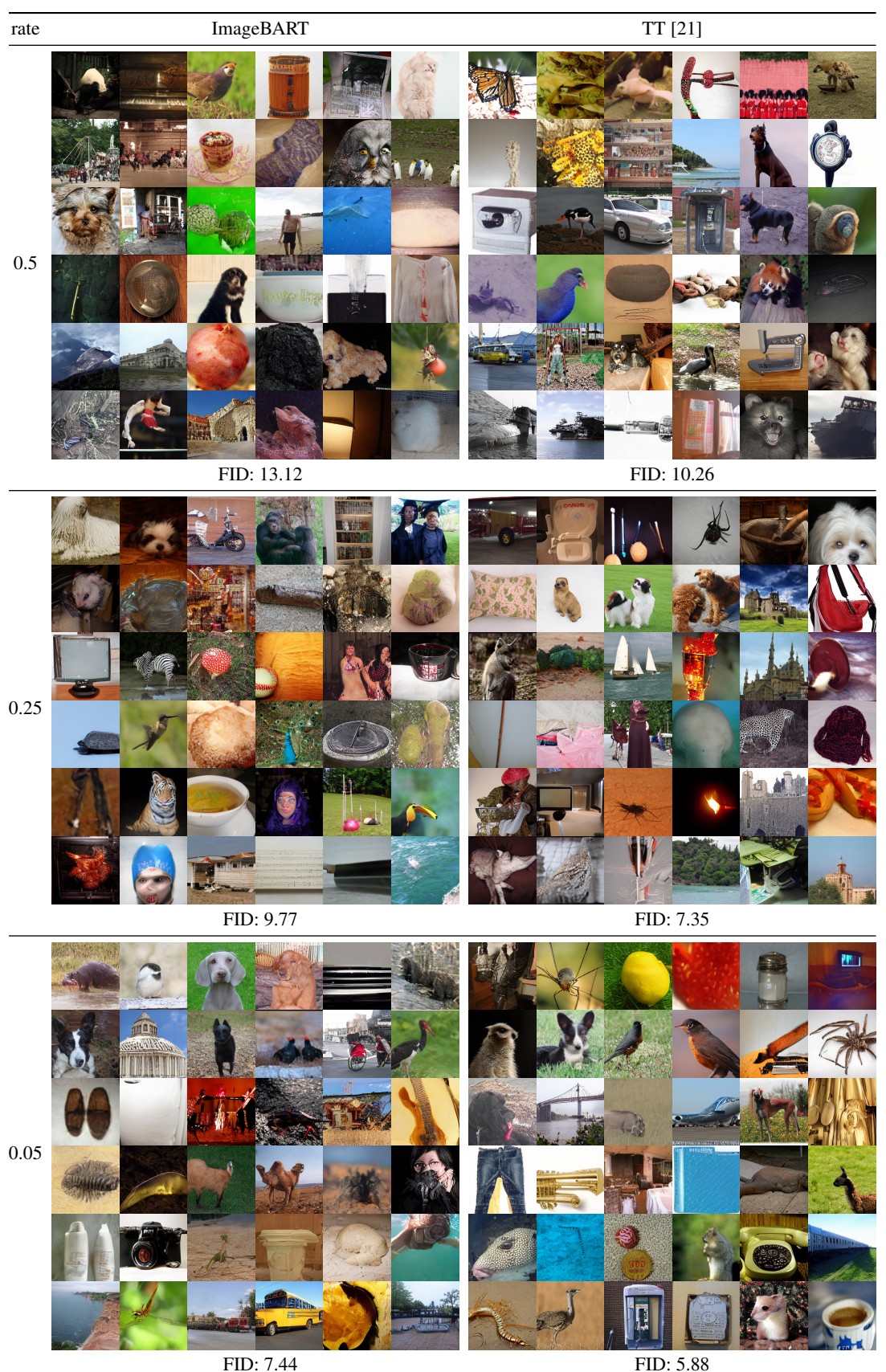

Figure 20: Qualitative and quantitative comparison of cIN samples for different rejection rates as in Tab. 1.

| *Sunset over the* | *Map of the world* | *Crowded scene* | *A small house* | *A photograph of a* | | *A vector illustration of* | |
| *skyline of a city.* | *in the year 2077.* | *in front of a pub.* | *in the wilderness.* | *beach.* | *crowd of people.* | *a tree.* | *the brain.* |

ImageBart

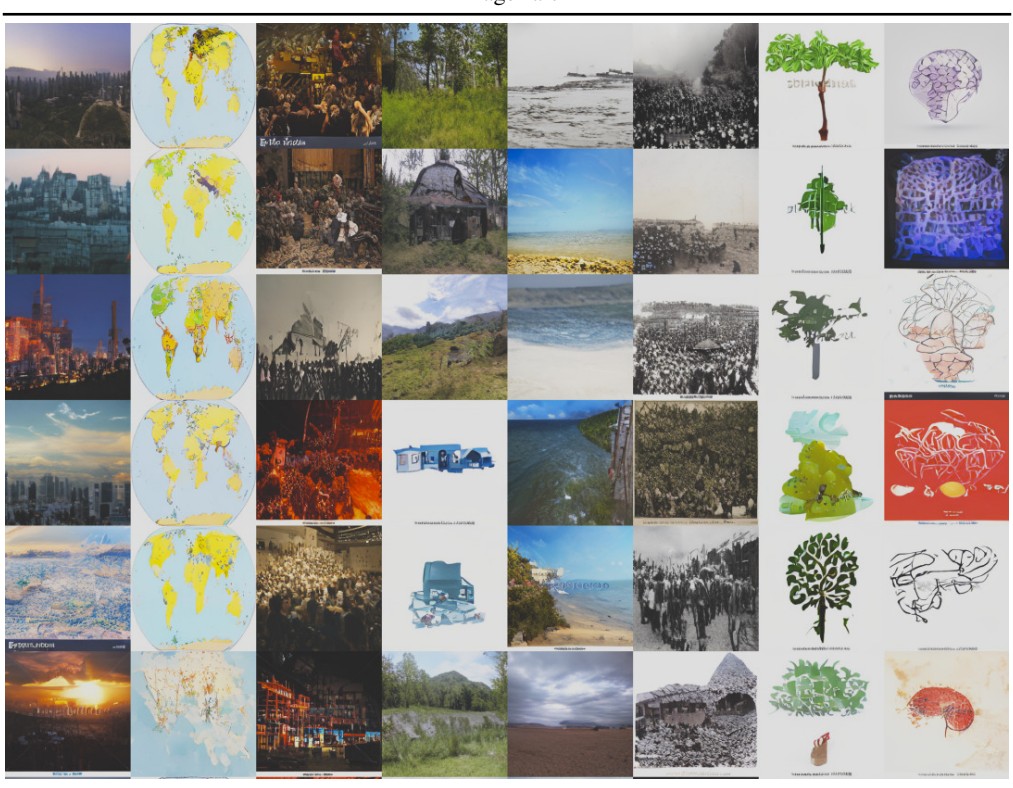

TT [21]

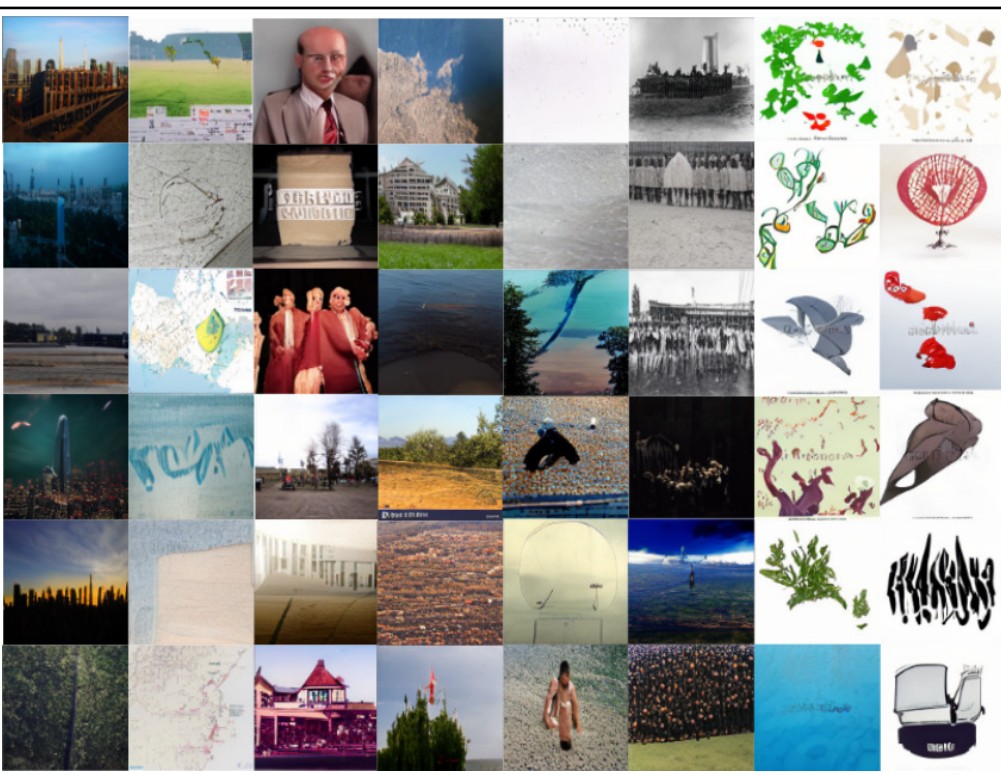

Figure 21: Random samples of text-conditional ImageBART and the text-conditional version of TT for the user defined text prompts above each row.

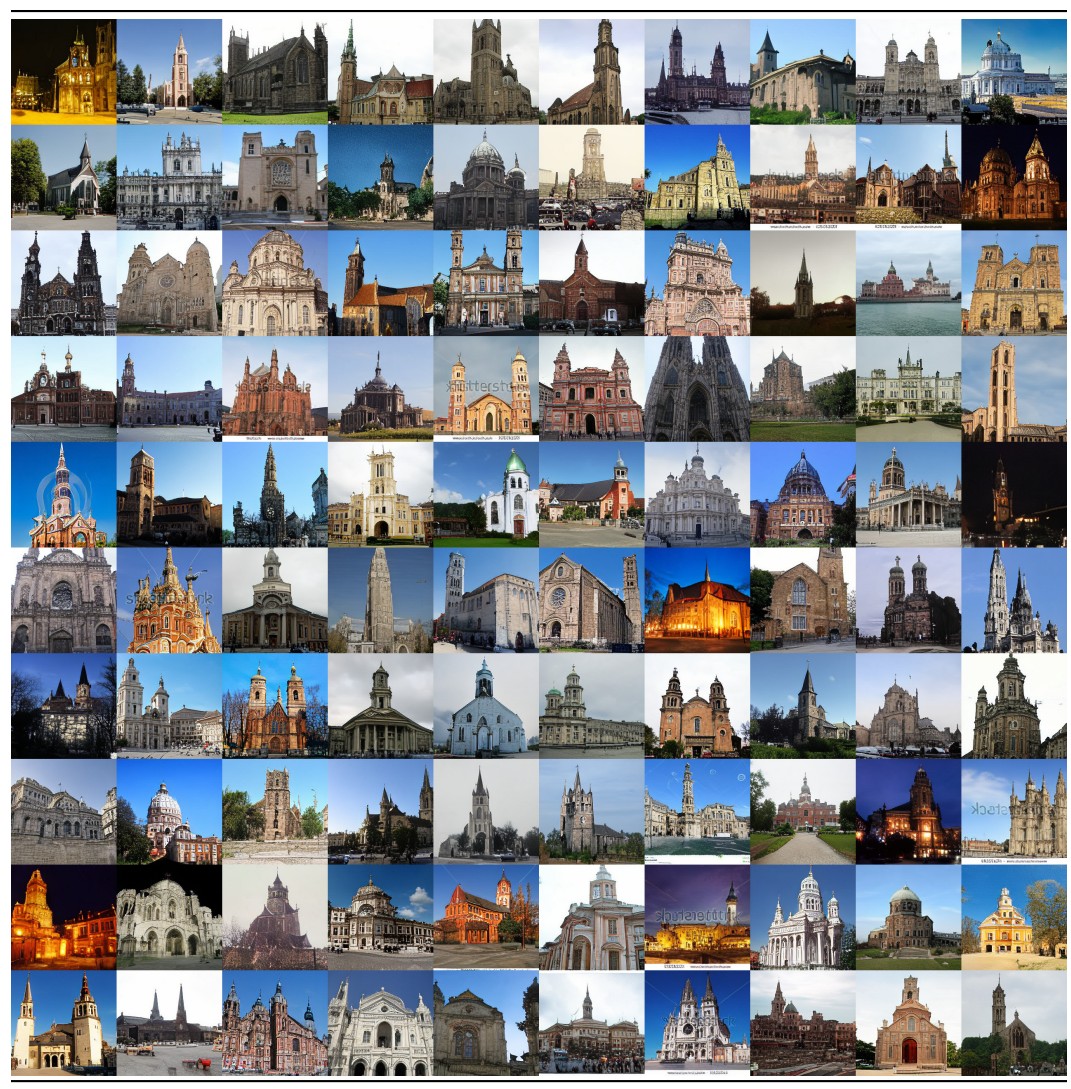

Figure 22: Additional $256 \times 256$ samples on the LSUN-church dataset.

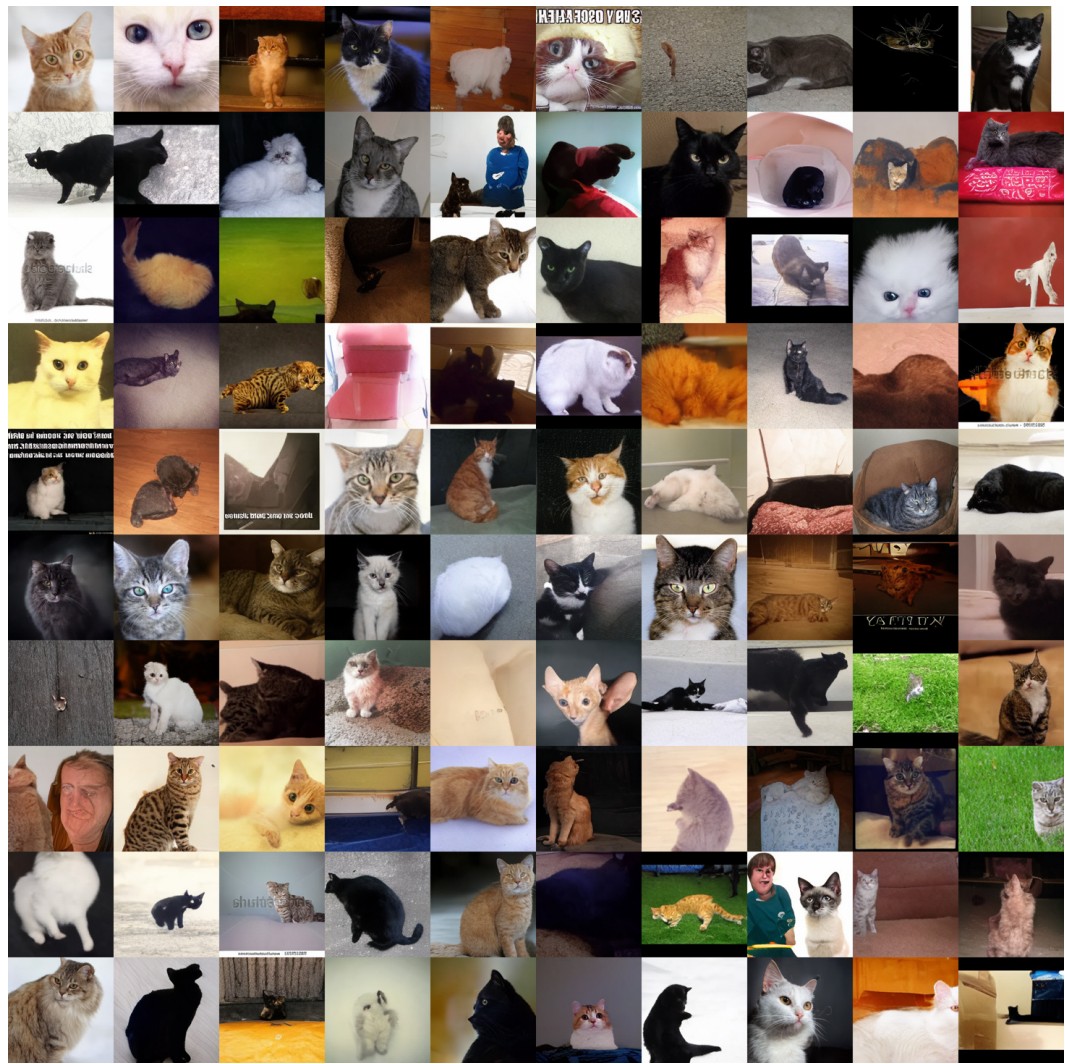

Figure 23: Additional random samples from our model trained on the LSUN-Cats dataset.

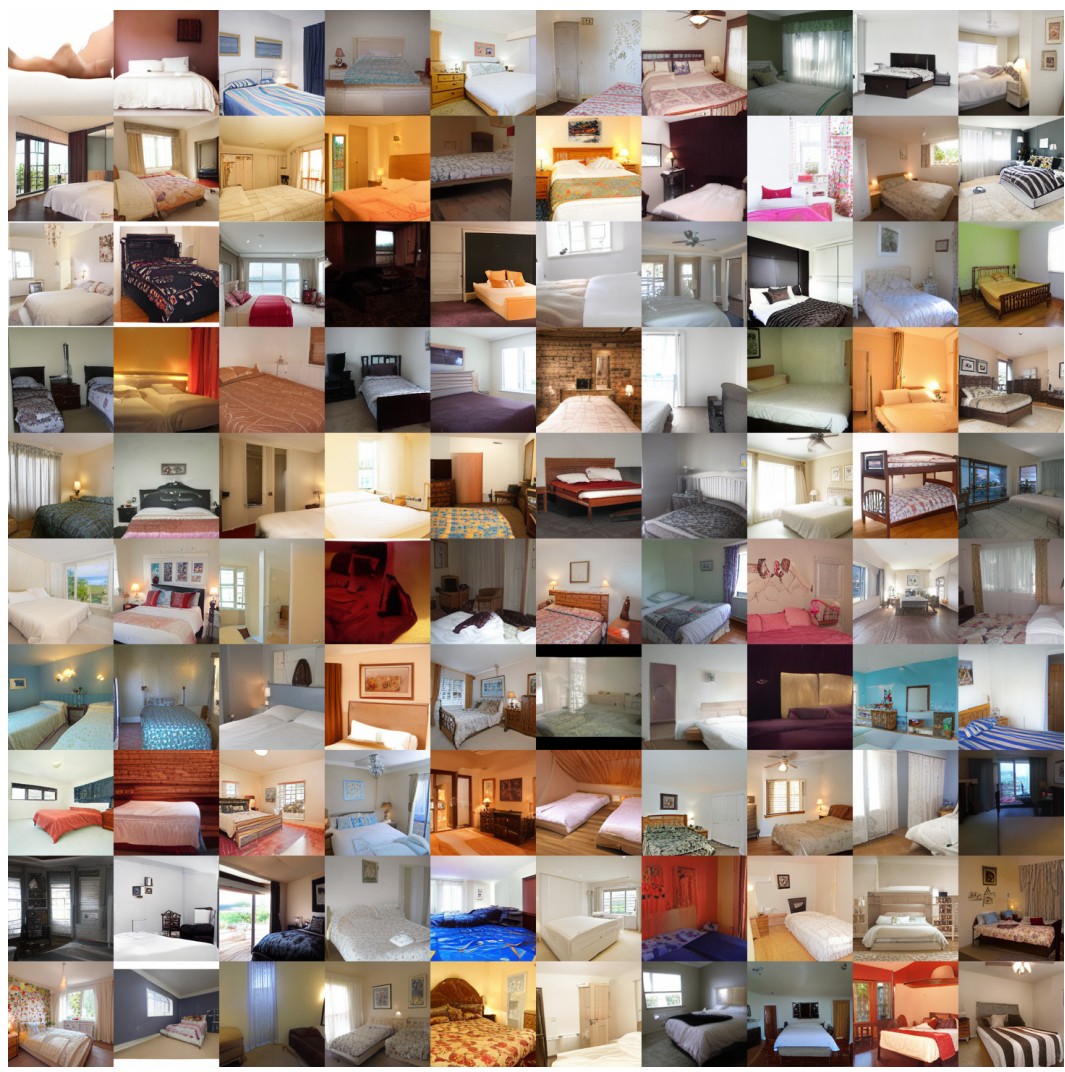

Figure 24: Additional random samples from our model trained on the LSUN-Bedrooms dataset.

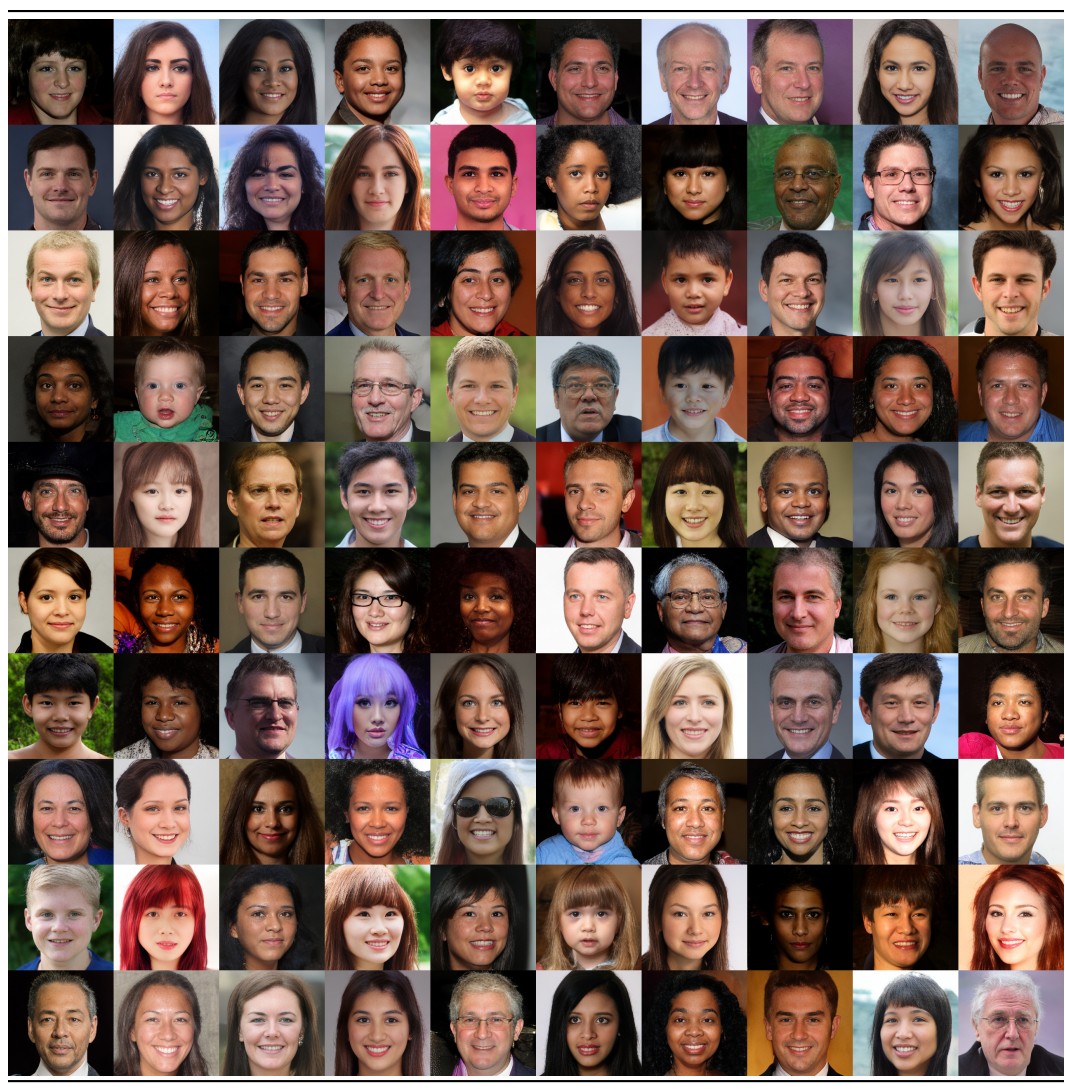

Figure 25: Additional $256 \times 256$ samples on the FFHQ dataset

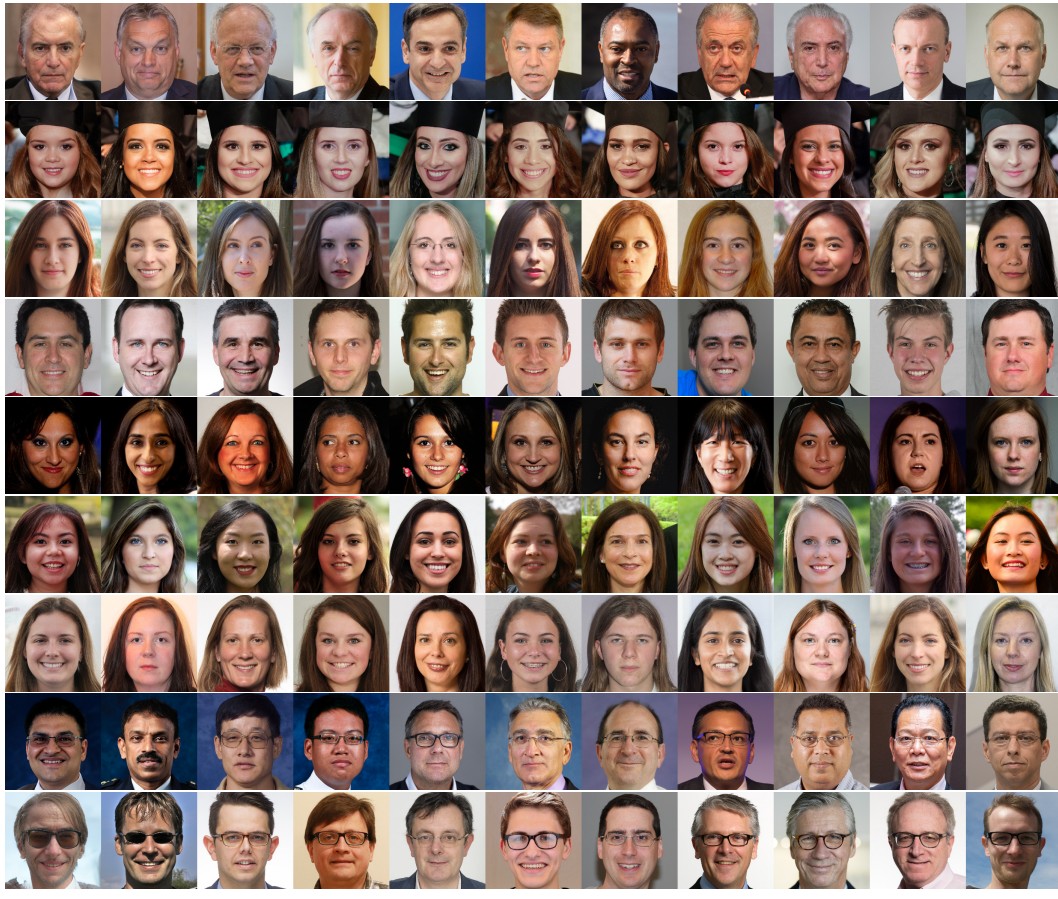

Figure 26: Nearest neighbors to samples from ImageBART from the FFHQ train set measured by averaging over different feature layers of a VGG-16 trained on ImageNet. The first example in each row shows a generated sample from our model. The remaining ones depict the corresponding nearest neighbors in ascending order.

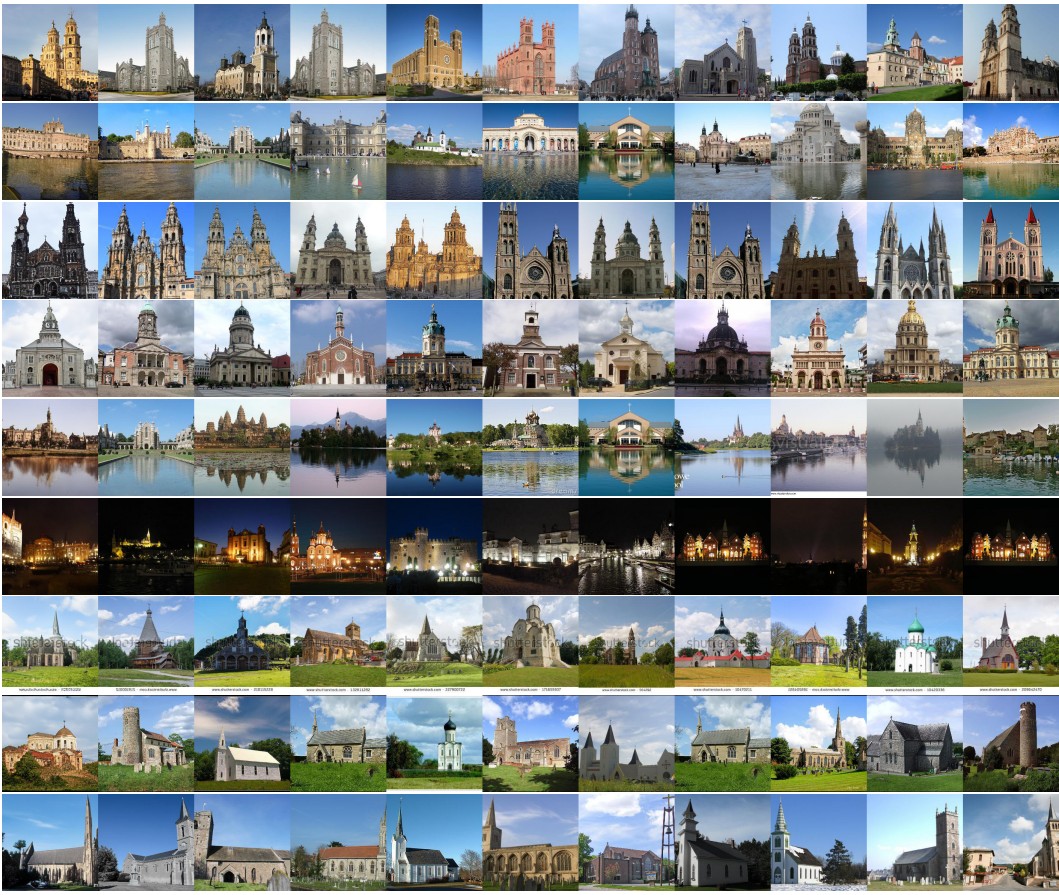

Figure 27: Nearest neighbors to samples from ImageBART from the LSUN-churches train set measured by averaging over different feature layers of a VGG-16 trained on ImageNet. The first example in each row shows a generated sample from our model. The remaining ones depict the corresponding nearest neighbors in ascending order.