# OpenReview forum: "ImageBART: Bidirectional Context with Multinomial Diffusion for Autoregressive Image Synthesis"
_NeurIPS.cc/2021/Conference — NeurIPS 2021 Poster_

### Official Review · Reviewer_9i6a · 2021-07-15

**Rating:** 6
**Confidence:** 3

**Summary:**

The paper proposed a coarse-to-fine image synthesis method which aims to incorporate global context into autoregressive models by a diffusion process. The authors divided the whole task into two subtasks: 1) discrete representation learning task 2) learning a Markov chain to reverse a fixed multinomial diffusion process. Experiment on unconditional image synthesis, class/text-conditional image synthesis and image synthesis with mask were shown.

**Limitations And Societal Impact:**

The authors provided a discussion on the energy impact and trade-off on sequence length and output quality. As indicated in the main review, the second part might need some experimental results and justifications.

**Main Review:**

## Originality

This work combines discrete image representation using VQGAN[1] with a multinomial diffusion process to introduce global context. The proposed solution of connecting those two ideas is intuitive and reasonable.

## Quality

I don't work on this very specific area, based on my understanding the paper is of good quality, but I just have the following the concerns and might change my rating if more clarifications are provided:
1. How is the bidirectional context incorporated in your method? The multinomial diffusion process is randomly introducing context switching, but there is no guarantee that which type of switching is being generated, please comment.
2. It seems that most of the experiment uses a diffusion process of T=3~6. How could you make sure that enough global context is being considered here? Also, does larger T always give better results? How do you choose T for a given dataset? I think an analysis on this would be helpful to justify the choice of T and show more insights.
3. For 4.3, I wonder how it would work for for TT [1] and your method if the input image is flopped horizontally and vertically (making the masked out region from the upper half to the lower half)? Would that produce different outputs?

## Clarity

1. The paper is overall well written with proper breakdown of each components of the method.
2. Line 36: Could you clarify why bidirectional context there does not provide valid factorization of the generative model density function?
3. The claimed contribution seems to mixing the concepts of bidirectional context, global context, hierarchical context all together. Therefore, making the relationships among them unclear. Please comment.

## Significance

Addressing unidirectional bias of AR models is an important problem. The proposed solution produces descent results while improving the sampling speed compared to previous SOTA methods.

[1] P. Esser, R. Rombach, and B. Ommer. Taming transformers for high-resolution image synthesis. *CoRR*, abs/2012.09841, 2020.

**Time Spent Reviewing:**

7

---

> ### Author Response · Authors · 2021-08-10
> **Response to 9i6a**
>
> __How is the bidirectional context incorporated in your method? The multinomial diffusion process is randomly introducing context switching, but there is no guarantee that which type of switching is being generated, please comment.__
>
> The diffusion processes makes sure that the markov chain, starting from the data representation, contains increasingly less informative representations, i.e. $MI(x_{0}, x_{t}) \geq MI(x_0, x_{t+1}) \geq ... \geq MI(x_0, x_{T}) = 0$
>
> __Also, does larger T always give better results?__
>
> This is a great question, which was also raised by other reviewers. To answer it, we performed an experiment on the FFHQ dataset where we fixed the total number of parameters and distributed it across 1,2,4 and 8 steps, respectively. See the "general response" for the analysis.
>
> __For 4.3, I wonder how it would work for for TT [1] and your method if the input image is flopped horizontally and vertically (making the masked out region from the upper half to the lower half)? Would that produce different outputs?__
>
> Interesting question. Yes, it produces different outputs, as the dataset (FFHQ) is heavily biased towards centered & aligned faces shown in a fixed pose. See also the figure at https://github.com/imagebart/neurips2021/blob/main/tt_vertical_flip_completion.pdf , where we vertically flipped the input image both in image and latent space and completed it with the TT baseline. We observe that the model tries to complete a lower half despite receiving a "flipped lower half", which we attribute to the fact that the dataset FFHQ contains only faces shown in an upright position.
>
> __Line 36: Could you clarify why bidirectional context there does not provide valid factorization of the generative model density function?__
>
> The chain rule of probability does not allow to factorize the full density in the bidirectional fashion without structural assumptions on the conditional independence of certain sequence elements. Thus, any bidirectional "BERT-style" model does not provide a normalized density and also renders sampling non-trivial. The latter is however an active area of research, see e.g. https://arxiv.org/abs/2105.14211 and https://arxiv.org/abs/1902.04094
>
> __The claimed contribution seems to mixing the concepts of bidirectional context, global context, hierarchical context all together. Therefore, making the relationships among them unclear. Please comment.__
>
> Thanks for pointing this out. In fact, all these terms describe the same property of our autoregressive model to attend to context from the previous scale,a (more) noisy but global image representation. Therefore, our entire model can be seen as a coarse-to-fine hierarchy, each stage of which is provided with global context from its predecessor (see also our answer to Reviewer 7J1S). From an NLP-perspective, this capability to attend to global context is equal to the ability to attend to context describing tokens to the right, often called "bidirectional context".  We clarified this in the revised version.

---

> > ### Comment · Reviewer_9i6a · 2021-08-26
> > **Raising Score to 6**
> >
> > Thank you for the response. I raise my score to 6 since the effect of the diffusion steps seems interesting.

---

### Official Review · Reviewer_7J1S · 2021-07-15

**Rating:** 6
**Confidence:** 4

**Summary:**

The paper addresses image generation problem and proposes a combination of 3 existing methods: VQ-VAE/GAN, multinomial diffusion and BART. The system works as follows: the images are first tokenized with VQ-VAE/GAN, then 3-5 steps of multinomial diffusion are run on discrete tokens, then those steps are inverted with BART at each step (starting from random noise). Quantitatively, paper evaluates FIDs on the LSUN-{Churches,Beds,Cats} and on the FFHQ datasets, choosing 6 methods as baselines (VDVAE, DDPM, StyleGAN2, BigGAN, DCT, TT). Qualitatively, same datasets + additional ImageNet results are presented. Also, the paper contains qualitative results for inpainting and/or text conditioning. A few ablations are presented: joint/independent training of diffusion steps, number of decoder/encoder layers.

**Ethical Concerns:**

No ethical issues with this paper.

**Limitations And Societal Impact:**

The authors adequately addressed the limitations and potential negative societal impact of their work.

**Main Review:**

The scores below are consistent with "Rating" legend.

## Originality = 6/10
The proposed method is new, although a combination of 3 well-known techniques: VQ-VAE/GAN, multinomial diffusion and BART. The main conceptual difference to prior work is usage of multinomial diffusion in VQ-space, BART seems like a particular implementation choice. Overall, the related work is reasonably cited, except for DALL-E paper - which is mentioned in results section but omitted from intro/related work. It might be desirable to add a concurrent work section and mention "Diffusion Models Beat GANs on Image Synthesis" paper (https://arxiv.org/abs/2105.05233) to give more context to the reader - but this is optional.

## Quality = 5/10
The submission is technically sound, however quantitative evaluation is quite limited:
1. While ImageNet samples are shown, FID is not provided.
2. An important ablation is missing: FID wrt number of diffusion steps, does it help to have more than 1? AR models are well-known to ignore learned latents, so what if just the last BART-diffusion step brings value?

## Clarity = 6/10
The submission is reasonably well-written. A few comments though:
1. [abstract, L4-6] "Not only is this unidirectional, sequential bias of attention unnatural for images as it disregards large parts of a scene until synthesis is almost complete." - This critique of AR models sounds strange to me. AR uses correct probability factorization, what is unnatural about it? Conversely, what should be considered natural? Also, what is meant by disregarding?
2. [abstract, L8-10] "we incorporate a coarse-to-fine hierarchy of context by combining the autoregressive formulation with a multinomial diffusion process" - Coarse-to-fine is mentioned multiple times in the paper but it's a somewhat questionable use of terminology. When I hear coarse-to-fine (and hierarchy in addition), I expect multiple spatial resolutions, but here it's always modeled at the same vq resolution, right?
3. [abstract, L14-15] "Experiments demonstrate the gain over current autoregressive models, continuous diffusion probabilistic models, and latent variable models" - please be more specific in what sense there is a gain: FID, speed etc.
4. [L33-34] "Imagine that you only have the lower half of an image and are looking for a completion of the upper half then these models fail at this minor variation of the completion task." - a natural answer would be to learn right-to-left AR. And they don't fail.
5. [L82] "Generative Models on Improved Representations" - please mention DALL-E. It's unpublished, but you do mention it in experimental section, so worth adding to related work.
6. [L141-142] "Note that the optimization of Theta via this objective includes the parameters of the learned codebook in addition to the parameters of the encoder and decoder" - I remember in the original VQ-VAE paper (https://arxiv.org/pdf/1711.00937.pdf) there was a problem with directly learning codebook because straight-through doesn't allow codebook updates, so it was instead learned with an additional VQ objective. But here you say codebook is learned with reconstruction objective. How do you circumvent the problem above? Perhaps to make the paper self-contained, it's worth to write out L_rec term in detail.
7. [Table 1] It would be great to mention number of parameters for all compared methods.
8. [Figure 2] Top row?
9. [L350-353] Duplicated citation.

## Significance = 5/10
Without addressing questions in "Quality", it's hard to judge significance:
1. ImageNet quantitative results are generally considered most trust-worthy, so comparing to representative methods there would boost confidence in this method.
2. How much this paper brings in terms of methods is conditioned on how much profit multiple steps of diffusion bring on top of AR applied to VQ.

**Time Spent Reviewing:**

4h

---

> ### Author Response · Authors · 2021-08-10
> **Response to 7J1S**
>
> __DALL-E and ADM__
>
> Thank you for this suggestion. We have added an discussion of these works to our related work section.
>
> __ImageNet FID__
>
> Please note that we provided a preliminary ImageNet FID Score in Fig. 10 of the supplementary. We also evaluated FID scores using different rejection rates ranging from 1 to 1/10, leading to scores from 20.01 to 7.98. We added the full study regarding different rejection rates to the main text.
>
> __An important ablation is missing: FID wrt number of diffusion steps, does it help to have more than 1? AR models are well-known to ignore learned latents, so what if just the last BART-diffusion step brings value?__
>
> To answer this question, we performed an experiment on the FFHQ dataset where we fixed the total number of parameters and distributed it across 1,2,4 and 8 steps, respectively. See the "general response" for the analysis. Regarding the problem of "ignored latents": In ImageBART, except for the compressed code, the latents are not learned, but fixed via the fixed diffusion process and the scales are trained independently.
>
> __AR uses correct probability factorization, what is unnatural about it?__, __right-to-left AR models__
>
> While any AR factorization is technically correct, the order of tokens induces an inductive bias (see also the supplementary material of TT) and also allows for different image modification abilities (e.g. lower half completion for raster-scan). To complete a different task (e.g. upper half completion), another model has to be trained (e.g. right-to-left). This means that a trained autoregressive model is restricted to the order it was trained on, whereas ImageBART is not, see Fig. 5 & 6 in the submission.
>
> __When I hear coarse-to-fine (and hierarchy in addition), I expect multiple spatial resolutions, but here it's always modeled at the same vq resolution, right?__
>
> It is correct that all scales of our model operate on the same, fixed VQ resolution. We clarified this in the revised version. However, as each stage of the forward diffusion process gradually removes information from the original discrete representation $x_1$, we found the attribute coarse-to-fine to be fitting in this case.
>
> __Quantitative Evaluation is quite limited__
>
> Following this comment, we have added additional quantitative results to the revised paper: We evaluated TT on LSUN Churches for unconditional image synthesis (FID 7.81 for TT and 7.32 for ImageBART, see Tab. 1) and Conceptual Captions for text conditional image synthesis (FID 28.68 for TT and 22.61 for ImageBART). We also performed a quantitative comparison between TT and ImageBART on the task of completing upper halfs on FFHQ, see "general response". There, we also report the results for unconditional image synthesis on FFHQ given an equal number of parameters and computational budget for TT and ImageBARTs with different number of scales and observe that more scales can lead to better results.
>
> __Number of Parameters__
>
> We added the number of parameters for all competing methods to the paper (ours can already be found in the supplementary).
>
> __Codebook Objective__
>
> Thanks for pointing this out. We use the same optimization scheme as presented in TT and clarified this in the revision.

---

> > ### Comment · Reviewer_7J1S · 2021-08-21
> > **Raising the score to 6**
> >
> > Thank you for conducting additional experiments and clarifications!

---

### Official Review · Reviewer_KT3X · 2021-07-16

**Rating:** 6
**Confidence:** 5

**Summary:**

This paper proposes ImageBART, an improved approach built on a recent success of an adversarial quantized representation learning model (e.g. VQGAN with autoregressive transformer) by replacing the single-scale autoregressive model with a multinomial diffusion process with a diffusion step-wise autoregressive encoder-decoder architecture. The (diffusion step-wise) encoder combined with a global cross attention mechanism from the adjacent diffusion step enables modeling of bidirectional context, which is not considered from the previous work (Taming transformers, TT).

**Limitations And Societal Impact:**

The limitations of this work is handled adequately in the supplementary material (appendix), focusing on 1. an energy requirement for training large-scale generative model, thereby advocating the use of the pre-trained high-performance compression model for downstream tasks , 2. the challenges of fitting the proposed models to small dataset due to a reduced inductive bias from the transformer architecture. I find that the arguments are valid and handled mostly well. However, perhaps it would be better to add an actual GPU runtime to train (or fine-tune) ImageBART for the main experiments to help the reader understand the scale of the computing requirements better.

**Main Review:**

Overall, the technical novelty feels somewhat incremental, as the proposed methods and the high visual fidelity results may mostly have stemmed from the high-performance compressed codebook with VQGAN. Otherwise, the proposed approaches (diffusion step-wise autoregressive models) would not have been feasible when operating on the pixel-level data space, rather than the compressed feature space. I'm aware that this is not a bad thing per se, so my assessment of this work is based on how well the writing and experiments are executed to demonstrate the clear advantage of the bidirectional context modeling compared to the original TT.

The overall writing, motivation and the justification of the methodological choices of each components are presented very well with enough materials for any potential audience on the generative models to catch up. Perhaps emphasizing that the autoregressive diffusion becomes feasible by operating on the compressed space to the "reverse diffusion models" section (in page 5) would make the description more complete & remove potential doubts.

The figures in table 1 convincingly demonstrated the potential failure modes of the current pixel-space diffusion models compared to ImageBART that leverages the powerful compression model. What feels to be desired, however, is more direct comparison of ImageBART against TT itself (to the Table 1 (right) and Figure 2, for example), because placing these comparisons for experiments other than the upper-half image completion (Figure 5) would make the improvements of ImageBART over TT more convincing & remove potential doubts on whether the higher quality samples came from the VQGAN itself or the additional multinomial diffusion of ImageBART. FIDs are already better than TT, and the "hierarchical generative model with bidirectional context" of ImageBART is expected to have more capability than just a "single-scale unidirectional autoregressive transformer" of TT. It would be nice if the authors could add more direct comparisons vs. TT to other qualitative results.

Similar impression goes to text-conditional image synthesis & local editing. While the advantage of the bidirectional context from is clear from Figure 5, the reader might ask whether the text-conditional generation was previously impossible with TT. I expect that the previous model can technically perform the same task, but the quality is expected to be worse, especially more so on the local image editing task, where the TT may suffer from degradation if the most upper-left parts are removed for the editing query. With the same logic, the difference between TT and ImageBART might be smaller for the text-conditional synthesis. If the authors have additional bandwidth, I would like to see whether this will be a correct implication of ImageBART on the demonstrated tasks. I understand that training additional transformer models require (huge) resources, but given the scale of the current experimental results, I expect that the authors may able to provide the material potentially in the revision.

Minor typos & questions

The inference speed of ImageBART is ~2x than TT. I expect that this may be from using fewer decoder layers (per step), but describing the main factor for the faster inference will be beneficial as one may naturally be curious about the cause, because ImageBART theoretically have more computational complexity (O(data_dim * num_difusion_step) compared to just O(data_dim) for TT).

Experiments 4.3 points the non-existent Fig. 4.3 (expected to be 5)?

The line spacing between 101 to 103 feels bit crammed

**Time Spent Reviewing:**

10

---

> ### Author Response · Authors · 2021-08-10
> **Response to KT3X**
>
> Thank you for the suggestion on emphasizing that the autoregressive diffusion becomes feasible by operating on the compressed space and for letting us know about the incorrect Figure reference and line spacing.
>
> __What feels to be desired, however, is more direct comparison of ImageBART against TT__, __text-conditional TT__
> To address a more direct comparison, we evaluated TT on LSUN Churches for unconditional image synthesis (FID 7.81 for TT and 7.32 for ImageBART, see Tab. 1) and Conceptual Captions for text conditional image synthesis (FID 28.68 for TT and 22.61 for ImageBART). We also performed a quantitative comparison between TT and ImageBART on the task of completing upper halfs on FFHQ, see "general response". There, we also report the results for unconditional image synthesis on FFHQ given an equal number of parameters and computational budget for TT and ImageBARTs with different number of scales and observe that more scales can lead to better results.
>
> __Inference Speed__
> Indeed, the main cause for this is the number of decoder layers. On each diffusion scale, the encoder layers only have to run once whereas the decoder layers have to run $n_\text{data\\_dim}$ times. This results in an approximate complexity of order $n_\text{scales} C(n_\text{encoder\\_layers}+ n_\text{data\\_dim} n_\text{decoder\\_layers})$, where $C$ is the complexity of a single transformer layer. The speedup from such an encoder-decoder transformer over a decoder only transformer with $n_\text{encoder\\_layers}+n_\text{decoder\\_layers}$ layers is therefore $\frac{n_\text{encoder\\_layers}+n_\text{decoder\\_layers}}{\frac{n_\text{encoder\\_layers}}{n_\text{data\\_dim}}+n_\text{decoder\\_layers}}$. We have added a description of this relationship to the paper.

---

### Official Review · Reviewer_TEXZ · 2021-07-19

**Rating:** 7
**Confidence:** 5

**Summary:**

This paper presents ImageBART, a generative model of images that uses multinomial diffusion to model the distribution of discrete latent codes extracted from a VQ-GAN encoder. Unlike prior diffusion based models, the reverse diffusion processes are modelled autoregressive, which enables a large reduction in the number of diffusion steps required. The model can condition on contexts such as a caption or semantic segmentation map, and unlike typical autoregressive models, is capable of image completion for arbitrary masked regions.

**Limitations And Societal Impact:**

The authors provide an extensive discussion of the limitations and potential negative impacts of the work in the supplementary material.

**Main Review:**

ImageBART is a fusion of the VQ-GAN introduced in Taming Transformers (TT, Esser et al, 2020), and the multinomial diffusion model introduced by Hoogeboom et al (2021). It aims to bring together the best of (increasingly effective) diffusion based generative modelling, and lossy discrete representation learning (VQ-VAE, VQ-GAN). ImageBART supports image completion with arbitrary masks unlike the autoregressive prior models used in TT.

Taming transformers is the most relevant baseline, given that it is similarly a likelihood based model trained in the same discrete latent space. I would argue that the comparison in Table 1 isn’t particularly valuable, given that the proposed models have a large number of parameters( >2B parameters for LSUN). Given neural scaling laws for autoregressive models (Kaplan et al, 2020), we know that performance and sampling quality is heavily influenced by model size and computational resources. These results may reflect these factors rather than the expressiveness / inductive biases of the underlying generative models. A more valuable experiment would compare ImageBART to TT using similar parameter counts / a fixed computational budget.

Also why not report ImageNet FID in the main text and compare it to other models (given that it is reported in Figure 10)?

It would be of interest to understand how the number of diffusion steps impacts model performance, beyond adding more parameters. One of the claims of the paper is that the hierarchical factorization of the data distribution is a helpful choice, but this isn’t experimentally verified other than through comparison to alternative works, where experimental conditions are different.

I find it strange to characterize the L1 loss as a likelihood, given that the autoencoder is trained adversarially, and the unnormalized density in equation (3) is likely to be degenerate. I think it’s fine to say: we’re doing lossy compression with a VQ-GAN, and learning a distribution of the compressed data.

[l257-259] “While our approach also models each transition autoregressively from the top-left to the bottom-right, the ability to attend to global context from the previous scale enables consistent completions of also the upper half” - I couldn’t figure out if the model can “see” context at the current scale when predicting masked pixels via the masked diffusion process, or if the visible context belongs to more coarse scales. If the former then that’s great, and if not a minor limitation is potentially not being able to match fine-grained detail in the masked / non-masked regions.

Conditional image generation results are qualitatively good but unsurprising: We have seen in a range of works that successful unconditional generative models can typically be conditioned on {image, text, other} context. The combination of masked-inpainting + caption-conditioning is a nice application.

Overall, I think this paper takes a natural step in combining two promising areas: diffusion-based generative models, and discrete representation learning. The proposed model offers benefits relative to pixel-level diffusion models or code-level AR models. The qualitative results are convincing, but due to confounding factors (e.g model sizes), it is hard to compare with baselines in an apples-to-apples way. The paper would benefit from some additional analysis of the benefits of using more / less diffusion steps.


**Time Spent Reviewing:**

3

---

> ### Author Response · Authors · 2021-08-10
> **Response to TEXZ**
>
> __It would be of interest to understand how the number of diffusion steps impacts model performance, beyond adding more parameters. / fair comparison with baselines__
>
> In order to address this question, we performed an ablation on FFHQ regarding the number of diffusion steps and report the results in the "general response" above ("On the Number of Diffusion Steps"). Furthermore, to enable a fair comparison, we fixed the number of parameters to 800M and distributed them equally across all scales and evaluated at the best available validation checkpoint.
>
> __L1 loss as a likelihood, ImageNet FIDs__
>
> Thank you for your suggestions. We note that this interpretation allows us to formulate the complete chain including the compression model as a concise framework but clarify that Eq. 3 is unnormalized. We have also added a study on ImageNet FIDs using different amounts of rejection rates (between 1 and 1/10) to the main text (resulting in scores between 20.01 and 7.98).
>
> __I couldn’t figure out if the model can “see” context at the current scale when predicting masked pixels via the masked diffusion process, or if the visible context belongs to more coarse scales. If the former then that’s great, and if not a minor limitation is potentially not being able to match fine-grained detail in the masked / non-masked regions.__
>
> Our model is provided context both from the previous scale and from the current scale. In each new scale, the first token is predicted only from the global context of the previous scale (see also yellow regions in Fig. 1) based on the corresponding masked sequence. Hereafter, the model gradually sees more context from the current scale, as it is additionally conditioned on the increasing number of already generated tokens (see also blue regions in Fig. 1) in an autoregressive fashion. Note that generated tokens in non-masked regions are always set to the visible context, thereby allowing the model to match the visible context also at the current scale (from the left).

---

### Author Response · Authors · 2021-08-10
**General Response**

We thank all reviewers for their time, helpful comments and for unanimously valuing the good results on image generation and modification. In particular, we are pleased that the reviewers find that the paper "is presented very well (R KT3X)'' "is technically sound and [...] well written" (R 7J1S), provides "convincing results" (R TEXZ),"is of good quality" (R 9i6a)'' and "takes a natural step in combining [...] diffusion-based generative models, and discrete representation learning (R TEXZ)''. We address the comments and questions below---thanks, we believe they add value to the work.
Note: This "general response" includes questions shared by multiple reviewers; however, we also address each of the individual comments in separate responses.

__On the Number of Diffusion Steps:__

We thank the reviewers for their remarks regarding the effect of varying the number of diffusion steps (denoted by $T$), and agree that this is worth an extended analysis. Hence, we perform an experiment for unconditional training on the FFHQ dataset, where we train a Taming Transformers (TT) baseline (corresponding to the case $T=2$ within our framework) with 800M parameters and three variants of ImageBART with $T=3$ (2x400M), $T=5$ (4x200M) and $T=9$ (8x100M), respectively. We assess both the pure synthesis and the modification ability of ImageBART by computing FID scores on samples and modified images (in the case of upper half completion as in Fig. 5), see the following tables.

__Tab. S1: Unconditional Image Generation FFHQ__

|           | FID ↓ | IS ↑          |
|--|--|--|
| TT        | 12.44 | 4.43 +/- 0.05  |
| ImageBART (T=3) | 12.55 | 3.98 +/- 0.07 |
| ImageBART (T=5) | 10.69 | 4.27 +/- 0.05 |
| ImageBART (T=9) | 10.81 | 4.49 +/- 0.05  |

__Tab. S2: Upper half completion FFHQ__

|           | FID ↓ | IS ↑          |
|--|--|--|
| TT        | 11.80 | 4.48 +/- 0.10 |
| ImageBART (T=3) | 9.25 | 4.49 +/- 0.13 |
| ImageBART (T=5) | 6.87 | 4.81 +/- 0.13 |
| ImageBART (T=9) | 6.64 | 4.86 +/- 0.15 |

In this experiment, increasing the number of diffusion steps leads to an improved visual quality of the synthesized samples (see Tab. S1). The main benefit of an increased number of diffusion steps is the improved controllability, which we quantify on the task of upper half completion in terms of FID (see Tab. S2). **Edit:** After convergence of the $T=9$ model, we observe that, for upper half completion, both FID and IS improve monotonically with $T$. For unconditional generation, FID seems to saturate after $T=5$ and IS initially drops below TT but improves upon it for $T=9$.

__Comparison to Taming Transformers (TT)__:

To address a more direct comparison, we also evaluated TT on LSUN Churches for unconditional image synthesis (FID 7.81 for TT and 7.32 for ImageBART, see Tab. 1 in the submission) and Conceptual Captions for text conditional image synthesis. For the latter, we trained a text conditional version of TT and evaluated FID and Inception Scores, as well as cosine similarity between CLIP embeddings of the text prompts and synthesized images to measure how well the image reflects the text:

| | FID ↓ | IS ↑ | CLIP ↑ |
|--|--|--|--|
| TT | 28.68 | 13.11 +- 0.43 | 0.20 +/- 0.03 |
| ImageBART | 22.61 | 15.27 +- 0.59 | 0.23 +/- 0.03 |

ImageBART performs better in all metrics while also enabling local image editing, which is not possible with TT. A (preliminary) qualitative comparison can be found at https://github.com/imagebart/neurips2021/blob/main/txt2img_imagebart_vs_tt.pdf

---

### Decision · Program_Chairs · 2021-09-27

**Decision:**

Accept (Poster)

**Comment:**

This paper presents several improvements to Taming Transformers through the use of a diffusion instead of autoregressive prior that enable more flexible conditional image generation and improved unconditional image generation and speed. Unlike prior work on diffusion that uses independent distributions for each layer, here autoregressive models for the reverse diffusion process are used which enables far fewer steps but results in slower sampling. The effectiveness of the approach is demonstrated in several conditional image generation tasks (e.g. arbitrary order inpainting, text-to-image generation, panorama generation). Reviewers were somewhat concerned of the novelty over Taming Transformers and the performance of the method on ImageNet. The authors addressed concerns with ImageNet results in the rebuttal, and provided more extensive comparison to TamingTransformers highlighting the utility of the diffusion process with autoregressive conditoinal distributions over the larger one step autoregressive model in that prior work. I'd argue for accepting this paper due to the interesting experimental results and applications, as well as the novelty in the diffusion generative model space of utilizing autoregressive decoders and showing it allows for far fewer steps.